# Integrase-mediated differentiation circuits improve evolutionary stability of burdensome and toxic functions in *E. coli*

Rory L. Williams [1,2] ✉ & Richard M. Murray [1]

Advances in synthetic biology, bioengineering, and computation allow us to rapidly and reliably program cells with increasingly complex and useful functions. However, because the functions we engineer cells to perform are typically burdensome to cell growth, they can be rapidly lost due to the processes of mutation and natural selection. Here, we show that a strategy of terminal differentiation improves the evolutionary stability of burdensome functions in a general manner by realizing a reproductive and metabolic division of labor. To implement this strategy, we develop a genetic differentiation circuit in *Escherichia coli* using unidirectional integrase-recombination. With terminal differentiation, differentiated cells uniquely express burdensome functions driven by the orthogonal T7 RNA polymerase, but their capacity to proliferate is limited to prevent the propagation of advantageous loss-of-function mutations that inevitably occur. We demonstrate computationally and experimentally that terminal differentiation increases duration and yield of high-burden expression and that its evolutionary stability can be improved with strategic redundancy. Further, we show this strategy can even be applied to toxic functions. Overall, this study provides an effective, generalizable approach for protecting burdensome engineered functions from evolutionary degradation.

As synthetic biology aims to engineer cells with the capacity to regulate and execute increasingly complex and burdensome functions, strategies which address the evolutionary instability of synthetic functions will only become more essential. It has long been observed that cell fitness negatively correlates with heterologous gene expression level[1], and increased burden results in a shorter evolutionary half-life of engineered functions[2,3]. Efforts to improve the evolutionary stability of engineered functions have taken a variety of forms[4], including the most straightforward goals of reducing mutation rate through sequence design[2,5] or host-strain engineering[5–7] to delay the appearance of mutations, or reducing burden constitutively[1,2,8] or dynamically[9] to mitigate their selective advantage. Additional strategies have improved evolutionary stability by various means of delaying or preventing the selection for mutations, including genomic integration of numerous copies[10,11], linking expression of a gene of interest (GOI) to an essential gene[2,12], addicting cells to the product of a metabolic pathway[13], or iteratively displacing populations of cells before mutational escape occurs[14]. While these strategies have indeed made headway, they vary in their ability to be generalized to diverse functions, and in the effort required to do so. Furthermore, each of these strategies requires the GOI to be expressed by every cell in the population, fundamentally restricting their application to GOIs that are non-toxic and of low burden.

A homogenous population of cells all performing the same function is largely unique to the laboratory environment, and recent years have seen the merit of breaking with this paradigm by engineering

[1]Division of Biology and Biological Engineering, California Institute of Technology, Pasadena, CA 91125, US. [2]Present address: Department of Biomedical Engineering, University of California Irvine, Irvine, CA 92697, US. ✉e-mail: rorylw@uci.edu

consortia instead of individual strains[15]. With inspiration from microbial communities, there have been several successful implementations of metabolic division of labor for producing biomolecules of interest[16–18]. This strategy has numerous advantages, including reducing the number of genes and associated metabolic load in each specialized cell type, allowing independent optimization of separate pathways, and spatially separating potentially incompatible functions. While these benefits may be realized by combining in co-culture independently engineered strains or species, additional attractive properties are possible with dynamically regulated division of labor within a single strain that encompasses both metabolic and reproductive functions. The use of differentiation to coordinate such division of labor is a recurring strategy used by microorganisms[19,20], but it has not yet been fully explored for addressing the evolutionary constraints of engineered functions. Natural systems that use differentiation to coordinate a division of labor are characterized by the cooperation of specialized cells carrying out distinct functions to realize an inclusive fitness benefit. This is seen with the multicellular cyanobacteria *Anabaena*, where photosynthetic vegetative cells reproduce and terminally differentiated heterocyst cells fix nitrogen and do not reproduce[21]. The importance of multicellularity for the evolutionary stability of differentiation-facilitated division of labor has been highlighted in both natural[21] and engineered systems[22], an observation which likely explains the absence of examples of differentiation in unicellular species. Multicellularity is necessary for natural differentiation systems because it minimizes the extent to which non-differentiating mutants can benefit from differentiated cells. However, while natural systems use differentiation to coordinate beneficial or essential functions, here we apply this strategy to functions that are instead both unnecessary for host survival and burdensome to cell growth. This important difference means that multicellularity would not bolster evolutionary stability in this context. This also necessarily means that there is no intrinsic safeguard against the expansion of non-differentiating mutants in our proposed system, a feature that we will later discuss and address.

The rational for implementing a differentiation strategy is to allow for a division of labor for the functions of (1) reproducing and (2) executing the function of interest, and critically to prevent the selection for mutations which disrupt this function. To adopt this strategy into a synthetic context, we develop a circuit architecture consisting of two cell types, with progenitor cells being specialized for the faithful replication of an encoded function in the absence of the burden associated with its expression, and differentiated cells for the execution of the encoded function (Fig. 1B). We utilize Bxb1 integrase-mediated recombination to simultaneously activate T7 RNAP-driven expression of a burdensome engineered function and inactivate the expression of π protein (an essential factor for R6K plasmid replication). We describe the strategy in which differentiated cells execute the function and can grow and divide indefinitely as differentiation. As differentiation results in loss of the R6K plasmid through dilution from cell growth and division, the proliferation of differentiated cells can be limited with antibiotic selection. We refer to the strategy in which the proliferation of differentiated cells is limited as terminal differentiation, similar to the use of this term in describing the terminally differentiated heterocyst cells of *Anabaena*[21]. Because T7 RNAP-driven expression is not activated prior to differentiation, there is no selective pressure for mutations which disrupt this function in the progenitor population. The limitation of differentiated cell growth with terminal differentiation then prevents such mutations from exponentially expanding in the differentiated cell population.

Here, we demonstrate computationally and experimentally that terminal differentiation increases the evolutionary stability of burdensome engineered functions, and that this strategy can be improved with strategic redundancy. We show further that terminal differentiation is robust both to mutations which relieve expression burden and to the level of burden. Finally, this robustness to burden is highlighted by the application of terminal differentiation for the production of a functional toxic protein.

## Results

### Terminal differentiation is a general strategy for addressing evolutionary stability

An ideal strategy for improving evolutionary stability is agnostic to the engineered function being expressed and can be readily implemented without requiring extensive specialization for each use case. As we will demonstrate computationally, terminal differentiation fulfills these criteria by being robust to both burden level and to mutations which disrupt the function of interest. We develop this intuition with cartoons and deterministic modeling by comparing the performance of differentiation architectures to the benchmark case of engineered expression in which every cell in a homogenous population both encodes and expresses the function. In this strategy, which we designate naive expression (Fig. 1A, C), the initial population of producer cells has a reduced growth rate due to the burden associated with the engineered function. Mutations which inactivate the expression of the burdensome engineered function, designated as burden mutations, give rise to non-producers which do not express the function and have a wild-type growth rate. Loss of expression of the function at the population level results from (1) mutations occurring during DNA replication which disrupt the encoded function, and (2) burden associated with the expression of the function providing selective pressure for these mutations during cell growth and division. Recognizing that reproduction and expression of the burdensome function must occur in same cells to select for non-producers, we proposed a strategy of terminal differentiation which segregates these two features through division of labor, thereby preventing evolution from destroying engineered functions (Fig. 1B).

Differentiation confines expression of the burdensome engineered function to the differentiated cell population by (1) activating expression through unidirectional differentiation, and terminal differentiation restricts replication and proliferation to the progenitor cell population by (2) limiting growth of differentiated cells. These two components are both necessary to complete the division of labor strategy. With differentiation, a population of non-producing progenitor cells have a wild-type growth rate and differentiate into producer cells which have a reduced growth rate (Fig. 1D). Burden mutations can occur in both differentiated producers (generating non-producers) and in progenitors (generating progenitors*), however, selective pressure for burden mutations only exists in differentiated cells as such mutations do not affect the growth rate of progenitors. Limiting the growth of differentiated cells as in terminal differentiation then fully prevents the expansion of non-producers (Fig. 1E).

The differentiation architecture, however, is susceptible to a new category of mutations occurring in progenitor cells which would destroy their capacity to differentiate. We refer to this category of mutations as differentiation mutations, and cells that can no longer differentiate as non-differentiators. While in the differentiation circuit without limited division, both non-producers and non-differentiators have a selective advantage, with terminal differentiation only non-differentiators have a selective advantage (Fig. 1E). We model differentiation, terminal differentiation, and naive expression deterministically with systems of ordinary differential equations describing carrying capacity limited growth in a chemostat with constant dilution. In these simulations, we model the production of an arbitrary protein by producer cells and examine the impact of (1) production burden through adjusting the specific growth rate of producer cells, and (2) the burden mutation rate. For differentiation, progenitor cells have a wild-type growth rate, the rates of differentiation and differentiation mutations are varied, and the number of post-differentiation cell divisions is modeled explicitly for terminal differentiation (Supplementary Note 1 for full description).

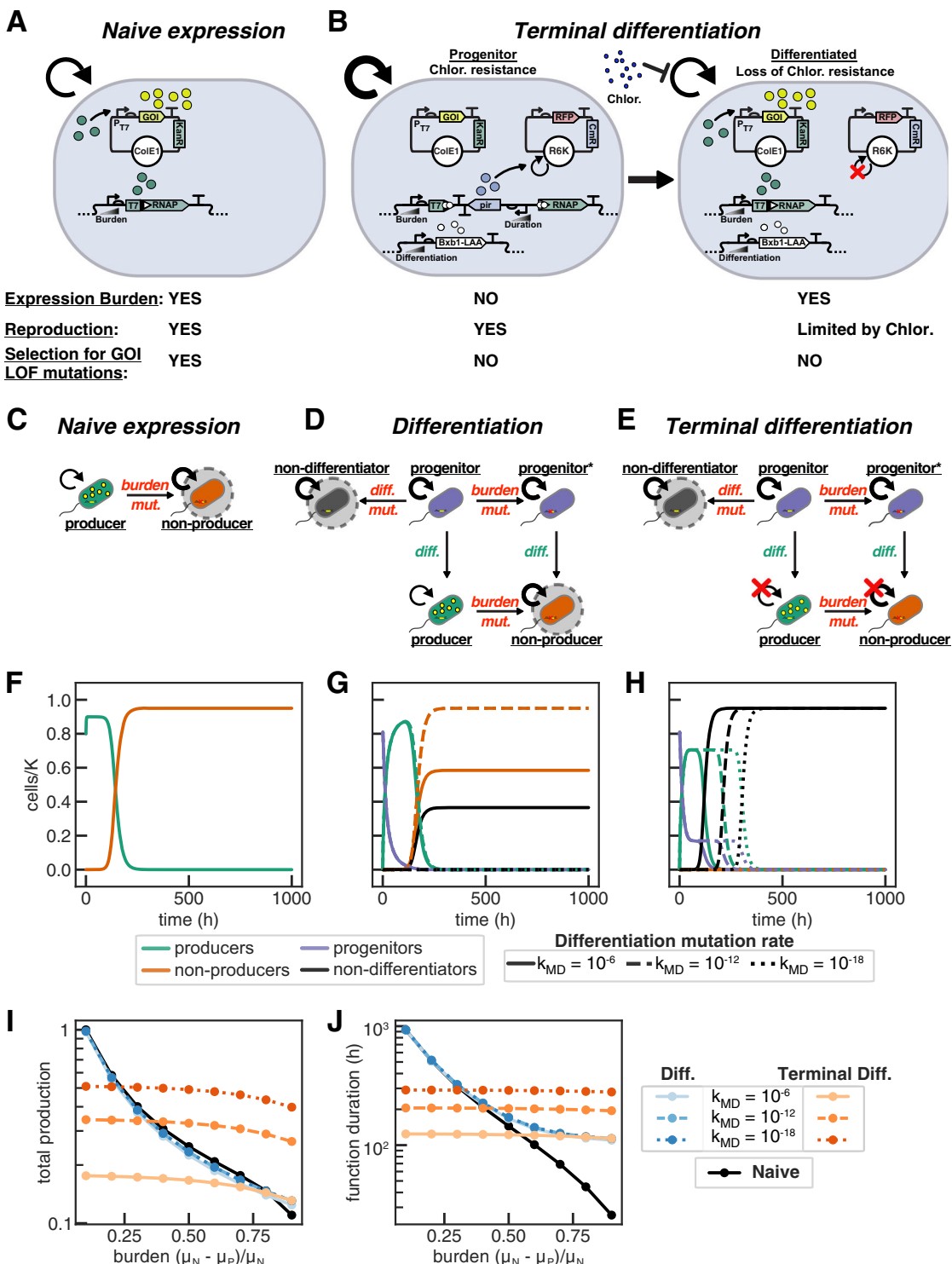

**Fig. 1 | Strategies for burdensome expression.** Experimental circuit design for (**A**) naive T7 RNAP-driven expression and (**B**) integrase-differentiation mediated activation of T7 RNAP which allows antibiotic selection against differentiated cells for terminal differentiation. **C**–**E** Cartoon depictions of naive expression (**C**), differentiation-activated expression (differentiation) (**D**), and differentiation-activated expression in which the number of cell divisions following differentiation is limited (terminal differentiation) (**E**). The thickness of the circular arrow indicates the relative growth rate, and states which are sinks are circled. **F**–**J** Deterministic ODE modeling of burdensome expression with circuits (**C**–**E**) growing with constant dilution and carrying capacity limited growth. $D = 0.1\,h^{-1}$, $\mu_N = 2\,h^{-1}$, $k_{MB} = 10^{-6} \times \mu = 10^{-6} \times \ln(2)/T_d$ ($T_d$ is doubling time, $\mu$ is growth rate which varies over time due to the effect of carrying capacity), $n_{div} = 4$, $K = 10^9$, initial cell population $8 \times 10^8$ cells, 1000 h duration, and varying differentiation mutation rate for differentiation and terminal differentiation (solid line: $k_{MD} = 10^{-6} \times \mu$; long-dashed: $k_{MD} = 10^{-12} \times \mu$; short-dashed: $k_{MD} = 10^{-18} \times \mu$). **F**–**H** Abundance of subpopulations plotted over time with burden due to production of 50% ($\mu_P = 1\,h^{-1}$). Producers (green) express the function and have a reduced growth rate, while all other cells (non-producers (orange), progenitors (purple), non-differentiators (black)) have a higher growth rate ($\mu_N$). **I** Total production and **J** duration of function (time at which production is 95% of max) plotted versus production burden for each strategy: Naive (black); differentiation (blues), terminal differentiation (reds), decreasing $k_{MD}$ light to dark).

Our intuition behind these architectures bears out in modeling, revealing terminal differentiation to be uniquely robust to burden mutations and to the degree of burden. Following what has long been observed experimentally, with naive expression non-producers overtake the population more quickly with higher burden expression, resulting in a faster loss of function (Fig. 1F, I, J; Supplementary Fig. 1D). While this is also true for differentiation without limited growth (Fig. 1G, I, J; Supplementary Fig. 1D), the duration of function for terminal differentiation, strikingly, is unaffected by burden (Fig. 1H–J; Supplementary Fig. 1D). This is because burden mutations do not provide a selective advantage, and non-producers do not expand in the population (Fig. 1H). While the performance of naive expression and the differentiation circuit without limited cell division are susceptible to burden mutations and, therefore, sensitive to the rate of burden mutations, terminal differentiation is robust to this rate (Supplementary Fig. 1G). Further, decreasing the rate of the differentiation mutation uniquely improves longevity for terminal differentiation (Fig. 1I–J). As differentiation mutations are the singular Achille's heel of the terminal differentiation architecture, any decrease in the rate or probability of these mutations will increase the duration of function. Because this circuit is agnostic to the specific function being expressed and robust to the rate of mutations which disrupt that function, reducing the probability of breaking the differentiation mechanism will improve the evolutionary stability of any function regardless of burden.

While the strategy of terminal differentiation is robust to burden level and burden mutations, this comes at the cost of reduced expression of the function of interest. This is because (1) the progenitor cell population does not express the function, and (2) limiting the growth of a cell after differentiation also limits the production achieved by its lineage. If the burden is low, the benefit does not overcome the cost and naive production outperforms terminal differentiation. We can make sense of this by understanding the source of selective pressure. With naive expression, the expression burden provides the selective pressure for burden mutations. With terminal differentiation, where differentiation mutations result in circuit failure, the selective pressure instead comes from the rate of differentiation. From the perspective of progenitor cells, differentiation is equivalent to death, and effectively acts to reduce the growth rate of the population. A consequence of this is that lower rates of differentiation allow for longer duration of function, though with a slower rate of production due to a smaller fraction of cells being differentiated producers (Supplementary Fig. 1E). In this tradeoff between production rate and duration of production with varying differentiation rates, an intermediate differentiation rate strikes a balance and achieves the most total production (Supplementary Fig. 1E). With sufficiently low expression burden, the selective pressure for differentiation mutations in terminal differentiation due to the rate of differentiation is greater than that for burden mutations in naive expression. The strategy of terminal differentiation, therefore, is expected to be beneficial only for functions that are sufficiently burdensome. Further, the magnitude of this benefit is expected to increase with the degree of burden, and with decreases in the rate of differentiation mutations.

## Development of an integrase-mediated terminal differentiation circuit

In order to experimentally implement the differentiation strategy, we required (1) burdensome expression to be fully off in the progenitor cell population, (2) irreversible and inducible activation of an arbitrary function through differentiation, and, in the case of terminal differentiation, (3) means of limiting the growth of differentiated cells. To accomplish this, we used Bxb1, a bacteriophage serine integrase which catalyzes unidirectional DNA recombination between specific sequences of DNA[23]. With strategic placement of integrase attachment sites on the genome, a single integrase-mediated recombination event

can simultaneously activate and inactivate the expression of the desired genes[24–26]. To reduce the impact of leaky Bxb1 expression we used the strong ssrA degradation tag LAA[27], and to allow tuning of Bxb1 expression and, therefore, differentiation rate we used the salicylate-inducible promoter $P_{SalTTC}$ and its cognate transcription factor NahR$^{AM}$[28]. In order to limit the capacity of differentiated cells to proliferate in the case of terminal differentiation, we take advantage of the reliance of R6K plasmid replication on the π-protein encoded by *pir*[29]. We used the 3OC12-HSL (Las-AHL) inducible promoter $P_{LasAM}$ and its cognate transcription factor LasR$^{AM}$ to control the expression of π-protein, and placed its expression cassette such that the recombination event results in its excision (Fig. 1B). The π-protein abundance and R6K plasmid copy number at the time of differentiation can therefore be tuned with Las-AHL. As the R6K plasmid encodes the sole source of chloramphenicol resistance (CmR), the induction level of π-protein sets the limit on number of divisions possible upon differentiation in the presence of chloramphenicol selection. In an initial evaluation of integrase differentiation, we demonstrated that the differentiation rate and R6K copy number could be controlled, and the fraction of cells in the progenitor and differentiated state tuned with a combination of chloramphenicol selection and Las-AHL/salicylate induction (Supplementary Fig. 2).

Both to allow any arbitrary function to be expressed and to prevent leaky expression of the function in progenitor cells, we selected T7 RNAP, an orthogonal RNA polymerase broadly used in synthetic biology and bioproduction[30], to be activated by this recombination. Importantly, T7 RNAP can then drive the expression of any protein or burdensome engineered function desired by the user. To allow the expression level and burden to be tuned, the evolved IPTG inducible promoter $P_{Tac}$ and associated transcription factor LacI$^{AM}$ were used to control the expression of T7 RNAP[28]. Recombination-activatable T7 RNAP was integrated in a single copy on the *E. coli* genome, and a high copy ColE1-AmpR plasmid with T7 RNAP-driven sfGFP was used to report T7 RNAP expression. Initial designs which contained a ribosomal binding site (RBS) adjacent to an intact T7 RNAP coding sequence prior to recombination (Supplementary Fig. 3A, B) displayed leaky sfGFP expression in progenitor cells above negative control lacking T7 RNAP (Supplementary Fig. 3F). To address this we relied on previous studies splitting T7 RNAP into functional domains to rationally choose a split site[31]. With this strategy, there is no potential for leaky expression of functional T7 RNAP prior to differentiation, and the full-length coding sequence that is generated upon recombination contains a 17 amino acid insertion from the attL site and additional bases inserted to conserve the reading frame (Fig. 1B, Supplementary Fig. 3C). In the absence of Bxb1 integrase, sfGFP production was equivalent to the control without T7 RNAP present, and induction of Bxb1 allowed high-level T7 RNAP-driven expression (Supplementary Fig. 3F).

In addition to the first three necessary criteria that have been fulfilled, the emergence of non-differentiators should be delayed in the terminal differentiation architecture by addressing the rate of differentiation mutations. From our initial deterministic modeling, we observed that decreasing the rate or probability of differentiation mutations improves the functional duration for terminal differentiation by delaying the emergence of non-differentiators. We reasoned that increasing the number of independent mutations required to break the differentiation mechanism would yield more significant improvements than decreasing the rate of mutations by sequence design. To this end, we envisioned an identical circuit design (Fig. 1B) that instead has two T7 RNAP differentiation cassettes. Importantly the recombination of a single cassette should both activate the function and enable limiting the growth of differentiated cells. If a second identical cassette was integrated, recombination of both cassettes would be required to cease replication of the R6K plasmid and allow limitation of growth through antibiotic selection. Therefore, a single mutation preventing the recombination of one cassette would be

sufficient to obviate antibiotic selection and allow burden mutations to expand. However, if each differentiation cassette encoded a unique half of the π-protein, a single recombination event would ablate the expression of functional π-protein and with it the replication of the R6K plasmid. In this case, two independent mutations would be required to generate non-differentiators and circumvent antibiotic selection, and burden mutations would have no opportunity to expand.

To accomplish this, we split the *pir* coding sequence and tagged the N- and C-terminal fragments with the N- and C terminal fragments of the Cfa intein[32], respectively, functionally screened for R6K plasmid replication, and identified a functional split site (Supplementary Fig. 4). Expression of the intein-tagged fragments allows R6K plasmid replication, and inactivation of either the N-terminal fragment or C-terminal' fragment through integrase-mediated recombination results in loss of the R6K plasmid. This split-π protein design was incorporated into a 2x differentiation circuit in which cells have two integrase-activatable T7 RNAP cassettes, with each encoding a separate intein-tagged π-protein fragment. Both the 2x differentiation strain and the corresponding 1x differentiation strain (having a single cassette encoding the full-length π-protein) were genomically integrated with two salicylate-inducible integrase (Bxb1-LAA) cassettes (Fig. 1A, B; Supplementary Fig. 5). In characterizing these circuits in ranging induction conditions, we noted that the R6K copy number and performance of the 2x differentiation circuit was more sensitive to the concentration of Las-AHL (controlling production of π-protein/split-π protein) than the 1x differentiation circuit (Supplementary Method 1; Supplementary Figs. 6–9), and selected 10 nM Las-AHL as the appropriate concentration for subsequent experiments.

The rate of differentiation is important as it determines the fraction of producer cells and, therefore, the rate of production, as well as the duration of function as we previously discussed. Accordingly, we characterized the dose-response of Bxb1 induction with salicylate on differentiation rate for both the 1x differentiation circuit and the 2x differentiation circuit, in both the non-terminal differentiation (-chlor) and terminal differentiation contexts (+chlor) at two burden levels (Supplementary Figs. 10–11). This revealed very little leaky differentiation for both circuits (-1% or less differentiated cells as determined by GFP + fraction with flow cytometry without salicylate) regardless of burden (IPTG induction) or chloramphenicol selection. We also observed a good dynamic range between 10 μM and 30 μM salicylate for both circuits, though generally, 2x differentiation was somewhat more sensitive than 1x differentiation to salicylate concentration. Specifically, after 8 h growth following a 1:50 dilution, 1x differentiation achieved -0.2%, -18%, -64%, and -94% differentiated producer cells (GFP + ) with 0, 10, 20, and 30 μM salicylate in the lower burden condition (10 μM IPTG), with similar though slightly lower percentages in the higher burden condition (-0.2%, -13%, -56%, and -90%); and 2x differentiation achieved -1%, -30%, -76%, and -97% in the lower burden case, and -1%, -43%, -74% and -97% in the higher burden condition (Supplementary Fig. 10).

In order to assess the benefit of the differentiation and terminal differentiation architectures, it was important to experimentally compare these to an equivalent naive circuit in which all cells were expressing the burdensome function. We, therefore, constructed 1x and 2x naive T7 RNAP expression strains in which the genomically integrated cassettes were identical to the sequence produced upon Bxb1 recombination (with the exception of not containing LasRᴬᴹ and NahRᴬᴹ, the transcription factors that were unnecessary for naive expression), and characterized the dose-response of IPTG induction on T7 RNAP-driven GFP expression and its associated impact on growth rate (Fig. 2C, Supplementary Fig. 5C, D, Supplementary Fig. 12). With this characterization, we see a dose-response between IPTG induction and both GFP production and burden as inferred through growth rate. For 1x naive there was a -11%, -31%, -50%, and -57% growth

penalty with 10, 20, 30, and 50 μM IPTG, respectively, which yielded -2×, -3.8×, -4.9×, and -6.5× the OD normalized GFP production relative to uninduced at 8 h. Similarly, for 2x naive there was a -27%, -54%, -64%, and -65% growth penalty, with -3.1×, -4.8×, -6×, and -7.6× the OD normalized GFP production relative uninduced 1x naive, respectively (Supplementary Fig. 12).

## Integrase-mediated differentiation circuits improve the evolutionary stability of burdensome T7 RNAP-driven functions

To assess the capacity of differentiation strategies to improve the evolutionary stability of burdensome functions, we performed long-term experiments with T7 RNAP-driven expression of a fluorescent protein, which here serves as a model burdensome engineered function. We compared the duration and total amount of production achieved in cells with one or two copies of inducible T7 RNAP (1x and 2x naive) to our single-cassette and two-cassette differentiation circuits (1x and 2x differentiation). Both the single-cassette differentiation and two-cassette differentiation strains have two copies of inducible integrase, and critically all components in the naive and differentiation circuits were genomically integrated, ensuring precise copy number control and preventing effects due to plasmid partitioning (Fig. 2A–C, summary of integration cassettes in Supplementary Table 1). All genomic insertions were verified by whole genome sequencing (Supplementary Data 1). Experimental comparison of differentiation with terminal differentiation required only including chloramphenicol in the medium in the case of terminal differentiation, as without antibiotic present differentiated cells would grow unhindered after losing the R6K plasmid. Inducer and antibiotic conditions were uniform throughout the duration of the experiment, with the degree of burden tuned with IPTG (P$_{Tac}$ T7 RNAP) and differentiation rate tuned with salicylate (P$_{SalTTC}$ Bxb1-LAA). Experiments were ran for a total of 16 consecutive batch growths with 50× dilutions into a total volume of 300 μL every 8 h, for a total of 128 h (-88 doublings).

In these long-term experiments, we observe both the benefit of redundancy, and the superiority of terminal differentiation particularly with higher burden T7 RNAP-driven expression. Comparing 1x naive to 2x naive expression, we clearly see the benefit of redundancy in both the duration of production and total production achieved (Fig. 2C, D; Supplementary Fig. 13 for plots of individual biological replicates). As each copy of inducible T7 RNAP is genomically integrated, each copy must independently mutate in order to fully disrupt its expression. The magnitude of this redundancy benefit is reduced in the higher burden case, however, with 2x naive yielding -2.2× that of 1x naive in the lower burden case (-221000 ± 31000 vs. -101000 ± 8000), and -1.6× that of 1x naive in the higher burden case (-101000 ± 30000 vs. -65000 ± 4000). For both 1x and 2x differentiation and terminal differentiation, the initial rate of GFP production increases with the differentiation rate (Fig. 2A, B; Supplementary Figs. 14–15 for plots of individual biological replicates). However, higher differentiation rates also lead to an earlier decline in production. These two counteracting features result in an intermediate differentiation rate yielding the most total sfGFP production (Fig. 2D). With 1x differentiation, the terminal differentiation condition had a moderate negative effect on total production for both the lower burden (-1.56× 1x naive expression for diff. vs. -1.37× for term. diff. with 15 μM sal/10 μM IPTG) and higher burden (-2.1× 1x naive expression for diff. vs. -1.89× for term. diff. with 15 μM sal/10 μM IPTG) conditions. With 2x differentiation, terminal differentiation had minimal benefit in the lower burden condition (-1.53× 1x naive expression for diff. w/ 15 μM sal/10 μM IPTG vs. -1.66× for term. diff. w/ 20 μM sal/10 μM IPTG), and a larger benefit in the higher burden condition (-2.17× 1x naive expression for diff. w/ 15 μM sal/50 μM IPTG vs. -2.86× for term. diff. w/ 20 μM sal/50 μM IPTG).

As expected, naive expression is more sensitive to burden level, and performs worse at higher burden in comparison to both differentiation and terminal differentiation (Fig. 2D). In interpreting the

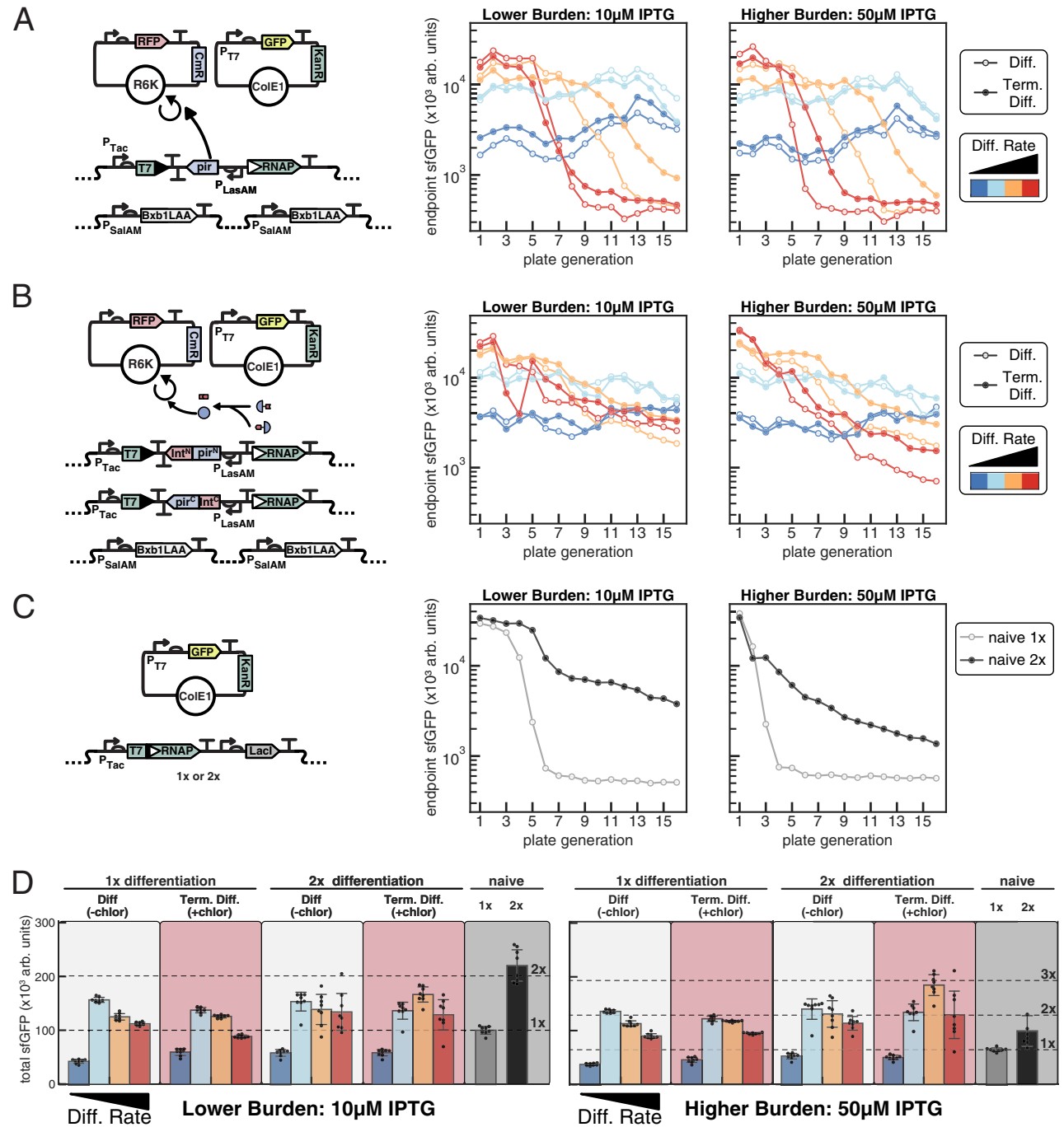

**Fig. 2 | Assessing the evolutionary stability of burdensome T7 RNAP driven expression from a high copy ColE1 KanR plasmid. A–C** 8 independent transformants were outgrown for 8 h in LB media with appropriate antibiotics and inducers before 50× dilution into experimental conditions. Cells were grown in 96 well plates in 0.3 mL, diluted 50× every 8 h for 16 total growths, and 50 µL endpoint samples taken to measure OD700, sfGFP (485/515 nm), and mScarlet (565/595 nm) fluorescence in 384 well Matriplates. Average endpoint sfGFP production ($n = 8$ biological replicates) plotted for each condition. **A, B** 1x differentiation and 2x differentiation. Each differentiation cassette additionally encodes NahR[AM], LasR[AM], and LacI[AM] (Supplementary Fig. 5). Cells were co-transformed with ColE1-KanR-$P_{T7}$ GFP and R6K-CmR-mScarlet and plated on LB + kan/chlor/30 nM Las-AHL. Colonies were outgrown in LB + kan/chlor/10 nM Las-AHL before 50× dilution into experimental conditions in LB kan with chlor (terminal differentiation, filled circles) or without chlor (open circles) with varying concentrations of salicylate (10, 15, 20, 30 µM: dark blue, light blue, orange, red, respectively) and IPTG (10, 50 µM). **C** Cells with one (1x naive, grey) or two (2x naive, black) copies of genomically integrated T7 RNAP were transformed with ColE1-KanR-$P_{T7}$ GFP, plated on LB + kan, and outgrown in LB + kan before dilution into experimental conditions. **D** Mean total cumulative production ± SD ($n = 8$ biological replicates) after 16 growths for all strains in all conditions shown in A-C. Dashed lines show 1×, 2×, 3× the average total GFP production of 1x naive. Source data are provided as a Source Data file.

effect of terminal differentiation for both 1x and 2x differentiation, we note that this has two opposing effects on the amount of production that will be achieved over the lifetime of circuit function. Terminal differentiation decreases output by limiting the growth of producer cells, but increases output by preventing the expansion of non-producers. Because these effects are opposing, one may dominate the other depending on the characteristic parameters. In the case of 1x differentiation, the negative effect dominates at both burden levels,

and terminal differentiation performs worse. With 2x differentiation, however, the positive effect dominates, particularly so at higher burden. The critical difference between 1x and 2x differentiation which explains this is that the 1x version requires only one differentiation mutation to yield non-differentiators, while the 2x version requires two mutations for the same effect. This delay in the emergence of non-differentiators provides additional time for the positive effect from suppression of non-producers to overcome the negative effect from limitation of producer growth.

In order to characterize the mechanisms which caused production of GFP to decrease or cease during the experiment, we performed selective plating assays from glycerol stocks saved after the last plate growth, and used Nanopore sequencing to identify causal mutations in a sample of colonies (Supplementary Fig. 16). We chose the higher burden (50 µM IPTG) and highest differentiation rate (30 µM salicylate) for this characterization as it was the condition which displayed the largest decrease in GFP production during the experiment. Select dilutions were plated from three of the eight replicates each for 1x/2x naive, 1x/2x differentiation, and 1x/2x terminal differentiation. Six colonies total for each strain across replicates were analyzed by sequencing. Sequences for the region of the ColE1 plasmid encoding T7 RNAP-driven GFP and all genomic integrations were obtained, with few exceptions due to failed PCR or insufficient reads. 1x and 2x naive had <1% producers in all replicates, as inferred through the fraction of GFP+ colonies. For 1x naive, sequencing identified causal mutations in the coding sequence of T7 RNAP at the P21 (T) locus in all six colonies, with 3 unique frameshift insertion mutations and 3 unique nonsense mutations, and no mutations were observed in the ColE1 plasmid. For 2x naive, however, no mutations were observed in the T7 RNAP coding sequence at the T or HK022 (H) locus, but instead mutations in the T7 promoter on the ColE1 plasmid were present in 4 of the 6 colonies (Supplementary Data 2). Mutations in the T7 promoter highlight the large contribution of transcriptional burden in this system[33], and the power of random plasmid partitioning in accelerating the fixation of mutations[34]. As well, that we see these promoter mutations in 2x naive but not 1x naive suggests the aggregate rate of generating and enriching for plasmid mutations through random plasmid partitioning is lower than the rate of genomic mutations which disrupt T7 RNAP expression, but higher than the rate of generating two such mutations. As we discuss the mutations identified in the differentiation and terminal differentiation circuits, we note that no mutations were observed on the ColE1 plasmid apart from those observed with 2x naive expression.

For 1x differentiation, 68–77% of colonies were non-differentiators, 23–32% were non-producers or non-differentiators that had lost the R6K plasmid, and none were producers or functional differentiators (mScarlet+, non-fluorescent, and GFP+, respectively, when plated on LB + Kan/Las/Sal). For 1x terminal differentiation, two replicates had 100% non-differentiators, and one replicate had ~99.8% non-differentiators and ~0.2% differentiators/producers (Supplementary Fig. 16). From our intuition and modeling, we know that while non-producers and non-differentiators have selective advantage with the differentiation architecture, only non-differentiators have selective advantage with terminal differentiation. The results of the plating assay reflect this, though the high abundance of non-differentiators in non-terminal differentiation suggests that differentiation mutations are more frequent. We sequenced 3 mScarlet+ non-differentiators, and 3 mScarlet-/GFP- colonies from 1x differentiation. All 3 mScarlet+ non-differentiators had identical inversions at the T locus, strongly suggesting that Bxb1 had catalyzed an inversion between the attB and attP sites rather than an excision. The sequence resulting from this recombination does not contain functional attB or attP sites, but does retain the capacity for π-protein expression. Of the three mScarlet- colonies, one contained a correctly recombined cassette that had a nonsense mutation in T7 RNAP (W221*), one had an inverted cassette

as described with intact *pir*, and one had a large deletion encompassing the attP right half through a portion of NahR^AM, also with intact *pir* (Supplementary Data 2). These data demonstrate that loss of the R6K plasmid occurs without loss of π-protein expression. As well, erroneous recombination by Bxb1, which results in a sequence that is inert to recombination, is likely more frequent than errors of replication having the same affect or solely disrupting the expression of T7 RNAP.

For 2x differentiation, ~36%, ~17%, and ~1% of colonies were differentiators/producers; ~64%, ~83%, and ~99% were non-producers or non-differentiators that had lost the R6K plasmid; and ~0.1%, <0.1%, and «0.1% were non-differentiators. For 2x terminal differentiation, ~74%, ~27%, and ~26% were differentiators/producers; ~21%, ~49%, ~71% were non-differentiators (mScarlet+); and ~6%, ~24%, and ~4% were non-producers or non-differentiators that had lost the R6K plasmid (Supplementary Fig. 16). Colonies that were mScarlet-/GFP- were only observed for 2x terminal differentiation when plating on media lacking chloramphenicol, and no such colonies were observed for 1x terminal differentiation. This suggests that the split-π protein design is more susceptible to stochastic loss of the R6K plasmid than the full-length *pir* design. Though non-differentiators were rare in the non-terminal differentiation condition, we sequenced 3 non-differentiators (mScarlet+) and 3 non-producers (mScarlet-/GFP-). In the non-producers, one colony had an inversion at the H locus as described previously with an intact *cfaC-pirR* fusion protein, and the cassette at the T locus had an inversion involving the integrase attachment sites but disrupting the *pirL-cfaN* fusion, ablating functional π-protein expression. The two additional colonies had this same T locus mutation, but high-quality sequences were not obtained for the H locus. For the non-differentiators sequenced, two had inversions maintaining expression of the split π-protein (one with failed sequencing at the H locus). The third non-differentiator sequenced had matching 108 bp deletions in both copies of the Bxb1 integrase, and both cassettes had not been recombined and did not have any mutations. For 2x terminal differentiation, we sequenced 2 colonies which were GFP+/mScarlet+, both of which had an inversion in the T cassette which maintained the intact *pirL-cfaN* fusion, and a correctly recombined H cassette with no mutations in the T7 RNAP coding sequence. The four non-differentiator colonies (GFP-/mScarlet+) colonies all had mutations which disrupted one or both integrase attachment sites at both the T and H loci but left π-protein expression intact. Simple inversions resulting from erroneous recombination were the most common, but a large deletion from the attP right half through a portion of T7 RNAP, and inversions involving a partial duplication of *pirL-cfaN* were also observed. (Supplementary Data 2).

Importantly, the plating and sequencing performed supports our hypothesis that the terminal differentiation strategy removes selective pressure for burden mutations. We also observed evidence that differentiation mutations occur and are selected sequentially in 2x terminal differentiation, as evidenced through the identification of cells in which one cassette was functional while the second had incurred a differentiation mutation preventing its recombination. It is likely that one such mutation reduces the aggregate rate of differentiation of the cell, and thereby provides a selective advantage. It is also apparent that errors in Bxb1-recombination are frequently the cause of mutations in both differentiation and terminal differentiation, and we speculate that these mutations occur at a higher rate than mutations due errors in DNA replication. This hypothesis is also supported by the experimental observation that 2x differentiation (-chlor) underperforms 2x naive in the low-burden condition (Fig. 2D).

## Terminal differentiation is uniquely robust to plasmid-based effects

As terminal differentiation is uniquely robust to burden mutations, we hypothesized that this robustness extends to other sources,

mutational or otherwise, which could impact expression. Because the GOI being expressed by T7 RNAP in these experiments is encoded on a high copy plasmid, plasmid mutations and copy number fluctuations (or even plasmid loss) can impact its expression without requiring mutation of the genomically encoded T7 RNAP. To investigate how such plasmid effects could influence each of our circuit architectures, we expanded the modeling framework previously discussed to reflect the experimental circuit design more accurately, model mutations stochastically, and incorporate plasmid-based effects. We explicitly modeled the genotype of each cassette, incorporated integrase expression cassettes, address integrase mutations and differentiation cassette mutations separately, and modeled the differentiation rate of each cassette as being linearly dependent on the number of non-mutated integrase expression cassettes. Both the total plasmid copy number and the fraction of plasmids with mutations likely can fluctuate widely due to plasmid partitioning effects. However, for simplicity and tractability, we considered plasmid mutation or plasmid loss to be binary, with a single stochastic event either mutating all plasmids, or resulting in complete plasmid loss. Furthermore, though antibiotic selection is used experimentally to ensure plasmid maintenance, communal resistance for certain antibiotics (e.g., β-lactams) can allow antibiotic-sensitive cells to persist in the presence of antibiotic[35]. To capture this effect, we modeled Michaelis-Menten antibiotic degradation by plasmid-containing cells and stochastic plasmid loss. While the growth of plasmid-containing cells was not affected by antibiotic concentration, the growth rate of cells that have lost the plasmid was determined with a Heaviside function. With this the growthrate is 0 if the antibiotic concentration is at or above the minimum inhibitory concentration (MIC), and is that of a non-producer if below the MIC. In simulating cells with one or two copies of inducible T7 RNAP in both the naive and differentiation cases, assumptions were required about the relative burden levels and production rates. In the case of differentiation, the producer growth rate ($\mu_P$) is the growth rate of a cell with one activated cassette of T7 RNAP, and the burden of the second copy produces a proportionate decrease in growth rate. For example, if a non-producer grows at rate $1\,h^{-1}$ ($\mu_N$) and a cell with one cassette active grows at rate $0.5\,h^{-1}$ ($\mu_P$), a cell with two cassettes active would grow at rate $0.25\,h^{-1}$. Production then was assumed to increase linearly with the decrease in growth rate (Supplementary Note 1 for full description of model implementation).

From these simulations, we recapitulate several observations from the initial deterministic modeling of the general strategies, and from our experiments. For both differentiation and terminal differentiation with one and two copies, we see that lower differentiation rates result in slower production that lasts longer, high differentiation rates yield faster production that breaks more quickly, and intermediate rates strike a balance and achieve the most total production (Fig. 3C, D). At low burden (higher $\mu_P$), terminal differentiation is counterproductive, but becomes beneficial as burden increases (Fig. 3C, D). We also see that naive expression performs comparatively well at low burden relative to high burden. For 2x naive expression we model both the case where one cassette alone yields the growth rate $\mu_P$ and its corresponding production rate (2x*), and where the two cassettes together yield the growth rate $\mu_P$ (2x). As expected, at low burden and high differentiation rate, 1x differentiation without growth limitation approximates the performance of both 1x and 2x naive, and 2x differentiation without growth limitation approximates the performance of 2x* (Fig. 3C–E). Further, the redundancy and mutational robustness provided with 2x differentiation improves performance relative to the one cassette case for both differentiation and terminal differentiation. Though we do not experimentally interrogate the impact of number of post-differentiation divisions ($n_{div}$) in the case of terminal differentiation, we do so computationally. While there is a benefit of increasing $n_{div}$ at low burden, this effect disappears at higher burden (Supplementary Fig. 17).

Critically, this stochastic modeling reveals that the robustness of terminal differentiation to burden mutations generalizes to plasmid-based mechanisms of relieving burden: both plasmid loss compensation by communal antibiotic degradation, and plasmid mutation. Incorporating stochastic plasmid loss in conjunction with antibiotic degradation and growth inhibition negatively affects performance of the naive and non-terminal differentiation architectures in a burden dependent manner, and this effect is much greater in the two-cassette case (Fig. 3C–E). The performance of the terminal differentiation architecture, however, is robust to this plasmid instability. We see this both in the total production achieved (Fig. 3C, D), and in tracking the population of cells which have lost the plasmid. When there is no antibiotic degradation, we see no accumulation of cells lacking the plasmid for any circuit, but with a sufficiently high rate of antibiotic degradation, we see a transitory rise in the fraction of cells that have lost the plasmid for naive (Supplementary Figs. 18–20) and differentiation architectures, but not for terminal differentiation (Supplementary Figs. 21–26). This effect increases with higher rates of antibiotic degradation, is more pronounced at intermediate (50%) burden than very low (10%) or very high (90%) burden for naive expression, and, in the case of differentiation, is influenced by both burden level and differentiation rate. As these plasmid-deficient cells are dependent on communal antibiotic degradation, they do not completely take over the population, but instead are eventually displaced by mutated cells which retain the plasmid.

Similarly, when we neglect antibiotic degradation and instead consider plasmid mutation, we see that increasing the rate of plasmid mutation negatively impacts naive and differentiation architectures, but not terminal differentiation (Fig. 3C–E). Further, that experimentally we observed mutations on the ColE1 plasmid disrupting T7 RNAP transcription in the 2x naive, but not 1x naive, corroborates simulation results where the impact of plasmid mutations is more pronounced for 2x than 1x naive expression. Though this effect is similar to plasmid loss in that it affects the redundant architectures more significantly, it is different in that it reveals its effect at lower burden (10%). As well, because there is no dependence on communal antibiotic degradation, the accumulation of cells with plasmid mutations is not transitory (Supplementary Figs. 18–26). Critically, the robustness of terminal differentiation circuits holds true when considering plasmid mutations, and we observe no selection for cells with mutated plasmids or any effect on production (Fig. 3, Supplementary Figs. 21–26).

While deliberately varying the rate of plasmid mutations experimentally is difficult, we may instead address plasmid instability through the choice of antibiotic resistance marker. Both to test this mechanism and to select any alternative selectable marker to use for the ColE1 plasmid, we used a plasmid identical to ColE1-KanR-$P_{T7}$-GFP which instead had ampicillin resistance (ColE1-AmpR-$P_{T7}$-GFP) and performed co-culture experiments of 1x naive and the parental strain JS006 transformed with these plasmids (Supplementary Fig. 27). As only 1x naive cells produce appreciable levels of GFP from these plasmids, we could observe that cells with only AmpR could allow cells with only KanR to grow in LB with both kanamycin and carbenicillin, while cells with only KanR did not allow cells with only AmpR to grow in the same condition. We therefore concluded that the choice of AmpR on the ColE1 plasmid would likely allow loss of expression through plasmid loss and shared antibiotic resistance, while the same mechanism would not hold, or would be much less apparent, with KanR.

Performing the same long-term evolution experiments described in Fig. 2 with the single modification of using AmpR marker instead of KanR corroborates the intuition we gained from modeling. With both the lower and higher burden conditions, changing the selectable marker largely removes the benefit from redundancy with naive expression (Fig. 3F, Supplementary Fig. 28). As well, while production is negatively affected by this change at higher burden for 2x

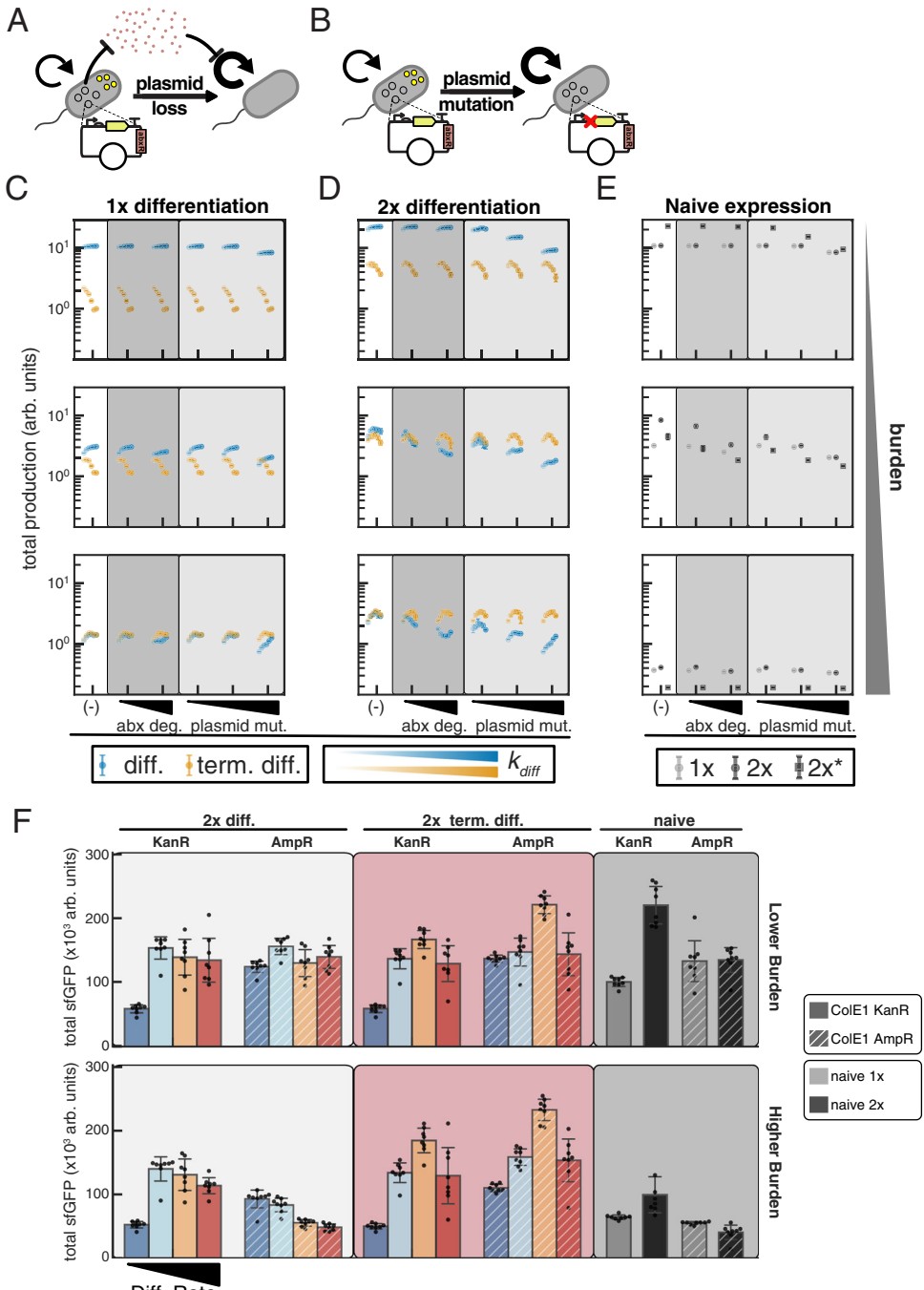

**Fig. 3 | Terminal differentiation is robust to plasmid effects which hinder naive and differentiation architectures. A** Schematic describing stochastic loss of burdensome plasmid which encodes an antibiotic resistance gene that allows for communal degradation. **B** Schematic describing stochastic plasmid mutation which relieves burden but does not impact antibiotic resistance. **C**–**E** Stochastic simulations of burdensome production in **(C)** 1x differentiation (blue) and terminal differentiation (orange), **(D)** 2x differentiation (blue) and terminal differentiation (orange), and **(E)** 1x (grey), 2x (black circles), and 2x* (black squares) naive architectures. Mean total production ± SD of 8 stochastic simulations of 20 consecutive batch growths with 50x dilutions. $\mu_P = 2\,h^{-1}$; 10, 50, 90 percent burden (increasing top to bottom); $K = 10^9$ cells; $k_{MB}$ (burden mutation) = $k_{MD}$ (differentiation mutation) = $k_{MI}$ (integrase mutation) = $10^{-6} \times \mu$; $k_{diff}$ = 0.2, 0.4, 0.6, 0.8, 1, 1.2 $h^{-1}$ (increasing left to right, light to dark shades); $n_{div} = 4$. Production rate and burdens/ growth rates as described in Supplementary Note 1. Naive 2x indicates a cell with two functional cassettes is described by the indicated burden level ($\mu_{PP} = \mu_P$; $\beta_{PP} = 1$), and growth and production rates of cells with one functional cassette ($\mu_{NP}$; $\beta_{NP}$) described by Eq. **1** and Eq. **2**. Naive 2x* indicates one functional cassette yields

the indicated burden ($\mu_{NP} = \mu_P$; $\beta_{NP} = 1$), with growth and production rates for cells with two functional cassettes described by Eq. **3** and Eq. **4**. Similarly, the growth/ production rates for 2x differentiation are equivalent to 1x naive when 1 cassette is activated, and 2x* naive when 2 cassettes are activated. Simulations with antibiotic degradation are with plasmid loss rate $k_{PL} = 10^{-4} \times \mu$; 100 µg/mL antibiotic; minimum inhibitory concentration MIC = 1.1 µg/mL; and antibiotic degradation, $2.52 \times 10^{-6}$, and $1.26 \times 10^{-5}$ µg/cell/h (left to right increasing abx deg). Simulations with plasmid mutation were modeled with $k_{PL} = 10^{-8}, 10^{-6}, 10^{-4} \times \mu$ (increasing left to right), 0 µg/mL antibiotic, and $V_{max} = 0$. **F** Comparison of total endpoint production after 16 plate generations for ColE1-KanR (solid) and ColE1 AmpR (diagonal stripes) variants of 2× differentiation (-chlor, light-gray shaded), 2× terminal differentiation (+chlor, red shaded), and 1x/2x naive (gray bars/black bars; gray shaded) with varying salicylate (10, 15, 20, 30 µM: dark blue, light blue, orange, red, respectively) and IPTG (10, 50 µM; lower, higher burden). Mean ± SD ($n = 8$ independent colonies) total endpoint GFP production plotted, with individual values plotted (black circles). Source data are provided as a Source Data file.

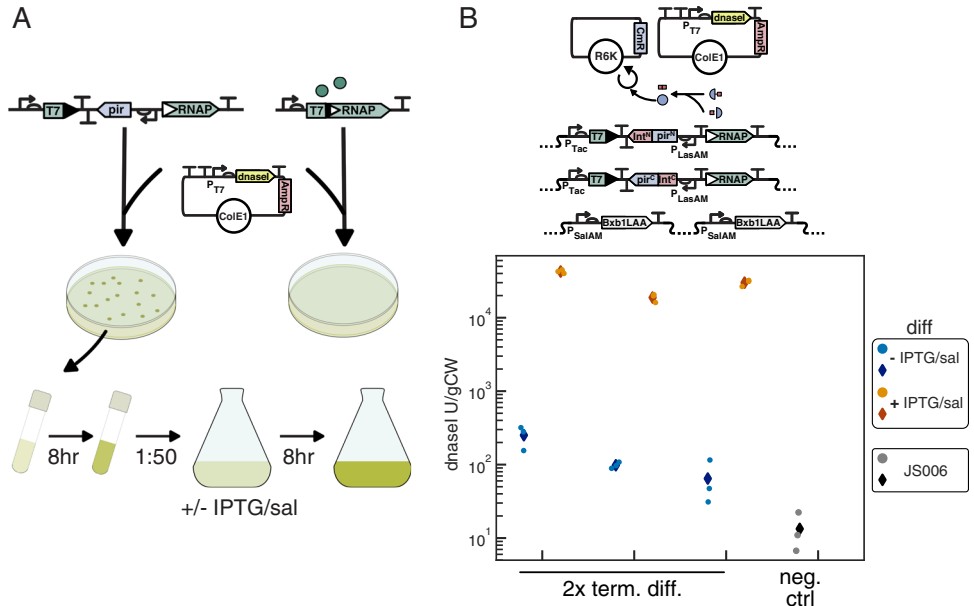

**Fig. 4 | Production of functional dnaseI using integrase mediated differentiation. A** Non-leaky T7 RNAP expression enables differentiation strains to replicate a plasmid encoding an insulated $P_{T7}$ dnaseI cassette, while cells with leaky T7 RNAP cannot. **B** For assaying dnaseI production, 2x differentiation cells were co-transformed with R6K-CmR-empty and Cole1-AmpR-T13m-T12m-$P_{T7}$-B0032-dnaseI-T7T and plated on LB + carb/chlor/30 nM Las-AHL. After 8 h outgrowth in LB + carb/chlor/10 nM Las-AHL at 37 °C, cultures were diluted 1:50 into 25 mL media with or without 10 μM IPTG/20 μM sal. Pellets for experimental cultures and JS006 negative control were harvested after 8 h, and lysate assayed for dnaseI activity (Supplementary Fig. 30). Lysates from three independent cultures for each condition, and one JS006 culture were assayed in triplicate. Three measurements (circles) plotted with average (diamonds) for induced (orange/red), uninduced (light blue/blue), and JS006 control (grey/black). Source data are provided as a Source Data file.

differentiation without limited growth, production is higher with AmpR than KanR for 2x terminal differentiation at both burden levels (Fig. 3F, Supplementary Fig. 28). While we cannot explain this performance benefit from our model, we speculate that the concentration of kanamycin used (50 μg/mL) can negatively affect expression even when KanR is expressed[36]. This stark contrast between terminal differentiation and differentiation or naive expression demonstrates that the robustness of terminal differentiation to burden mutations extends generally to plasmid-based mechanisms of reducing burden.

## Differentiation enables the expression of toxic functions

As the terminal differentiation architecture is robust to the level of burden associated with the function of interest, this suggests that even toxic functions could be expressed and made evolutionarily stable. To test this, we aimed to demonstrate that the differentiation circuit we developed could allow the production of a toxic protein that will result in cell death and chose dnaseI as a proof-of-concept example. As progenitor cells do not produce any T7 RNAP, we reasoned that a T7 RNAP-driven *dnaseI* would not be expressed in the progenitor cells, allowing the encoded function to be replicated without toxicity or selective pressure for mutations. However, construction of a dnaseI expression plasmid identical to that of sfGFP yielded only mutated plasmids. Characterization of leaky expression from a $P_{T7}$-GFP plasmid in the absence of T7 RNAP revealed fluorescence above background, explaining this inability to isolate functional plasmids (Supplementary Fig. 29). Incorporating two insulating terminators upstream of the T7 promoter mitigated leaky expression in the absence of T7 RNAP (Supplementary Fig. 29), and this insulation in conjunction with reducing the RBS strength allowed construction and isolation of a correctly sequenced dnaseI expression construct. While leak could have also been reduced by using the T7/lacO promoter and an additional source of LacI on the expression plasmid, this would not eliminate leaky expression in progenitor cells upon induction of differentiation and T7 RNAP. Highlighting the importance of

preventing leaky expression of toxic functions, transformation of 1x differentiation and 2x differentiation cells with insulated dnaseI plasmid yielded ~600 cfu and ~1000 cfu, respectively, while 1x naive and 2x naive strains yielded 1 and 0 colonies, respectively, compared to >10⁴ cfu for both when transformed with ColE1-AmpR-$P_{T7}$-GFP control (Fig. 4, Supplementary Table 2).

Using the insulated T7 RNAP-driven dnaseI construct, we demonstrated that differentiation could enable expression of this toxic product. To assess the capacity of differentiation to enable functional dnaseI expression, we co-transformed the 2x differentiation strain with the insulated dnaseI expression plasmid, and an R6K-CmR-empty plasmid. After outgrowth without induction, cells were diluted into 25 mL cultures with or without induction with 20 μM salicylate and 10 μM IPTG. After 8 h of growth, un-induced cultures grew to cell densities equivalent to JS006 control (7.2–7.6 g wet weight per liter vs. 7.6 ctrl), while induced cultures reached lower cell densities (2–4.4 gCW/L). Diluting 1:50 into fresh media yielded similar densities for the uninduced cultures after 8 h growth (7.2–7.6 gCW/L), while induced cultures had grown minimally after 8 h (OD600 < 0.05) and to a range of densities after 16 h total growth (2–6.8 gCW/L). The growth of the induced and uninduced cultures indicates that the dnaseI plasmid minimally affected growth when T7 RNAP is not expressed, and that 20 μM salicylate induction likely results in more complete differentiation in a large shaking culture in comparison to small volumes in 96 well microplates in previous experiments. Expression of functional dnaseI was quantified with an activity-based assay on lysate extracted from cell pellets (Supplementary Fig. 30), with activity measured equivalent to ~1.9–4.2 × 10⁴ U/gCW (~3.7 × 10⁴–1.9 × 10⁵ U/L) in the three independent induced cultures and ~65–250 U/gCW (~500–1800 U/L) in the uninduced cultures, compared to ~13 U/gCW (~100 U/L) for the JS006 negative control (Fig. 4B). This yield of dnaseI is on the same order of magnitude as yields reported using T7 RNAP to drive the expression of recombinant dnaseI using the LacI repressible T7 promoter in Bl21(DE3)[pLysS] (1.5 × 10⁴ U/L) and Bl21(DE3)[pLysE] (7.5 × 10⁴ U/L)[37].

## Discussion

Here we have developed architectures for implementing differentiation and terminal differentiation in *E. coli* for the expression of burdensome T7 RNAP-driven functions. Importantly, in our circuit design progenitor cells do not have an intact coding sequence for T7 RNAP, completely eliminating leaky T7 RNAP expression prior to differentiation. We computationally demonstrated that limiting the growth of differentiated cells with terminal differentiation provides robustness to burden level and burden mutations. We further demonstrated that reducing the rate of differentiation mutations delays the emergence of non-differentiators, thereby improving the evolutionary stability of the terminal differentiation architecture and with it any arbitrary function.

Experimentally, we developed a differentiation-activated T7 RNAP architecture in which all circuit components that could mutate to disrupt the process of differentiation or expression of T7 RNAP were integrated on the genome, both ensuring exact copy number control and preventing plasmid partitioning effects from accelerating circuit failure[34]. With the goal of delaying the emergence of non-differentiators and thereby increasing the longevity of the terminal differentiation architecture, we developed a split-π protein system using split-inteins[32]. In long-term experiments with repeated dilutions of independent cultures, we compared the performance of 1x and 2x naive T7 RNAP-driven expression to 1x and 2x differentiation and terminal differentiation. We demonstrated that the rate, duration, and total amount of production could be tuned by varying the differentiation rate, with lower differentiation rates enabling longer duration expression but at a slower rate. Differentiation was particularly beneficial in comparison to naive expression with higher burden as expected from modeling, and the redundancy and robustness to differentiation mutations provided by incorporating the split-π protein design into the terminal differentiation architecture proved effective. Though here we demonstrate the effectiveness of terminal differentiation and the stability benefit gained from requiring two mutations instead of one to generate non-differentiators, this can be viewed as demonstration of the power that redundancy can provide in synthetic biology. In considering the scaling of this strategy to longer times and larger population sizes, we expect that higher degree of redundancy will be necessary. As well, addressing errors of Bxb1 recombination which serve as a dominant source of differentiation mutations may improve the performance of terminal differentiation at each level of redundancy. Scaling this specific architecture through further splitting of the π protein may be infeasible, but the toolkit of synthetic biology certainly has means of allowing this strategy to scale further, through the inactivation of essential genes, activation of toxins, or otherwise.

We demonstrated both computationally and experimentally that effects due to instability of the T7 RNAP-driven expression plasmid and communal antibiotic resistance can negatively affect the performance of naive expression and differentiation, but that terminal differentiation circuits are robust to this effect. We further showed computationally that the robustness of terminal differentiation circuits to burden mutations extends to the general case of plasmid mutations which disrupt the function of interest. Though genomic integration of functions is more time consuming and cumbersome than plasmid transformation, plasmid instability and cost considerations for antibiotics and inducers in large cultures have often made genomic integration of constitutively expressed functions the preferred method for bioproduction in industry[10,38]. However, terminal differentiation mitigates effects of plasmid instability, potentially allowing the stability benefits of genomic integration to be obtained with the ease of plasmid transformation.

Finally, because there should be no limit on the degree of burden or toxicity of a function expressed with our differentiation system (so long as the toxicity is limited to the cells expressing the function), as a proof of concept we demonstrated that differentiation could enable the expression of functional dnaseI. In the course of this demonstration, we discovered that in the absence of leaky expression of T7 RNAP, non-T7 RNAP sources of leak could prevent isolation of correctly assembled dnaseI expression plasmids. While we mitigated this problem through the incorporation of insulating terminators to prevent transcriptional read-through from upstream of the T7 promoter, reducing the strength of the RBS was still required to isolate correctly sequenced plasmid. Improving this insulation and/or reducing any leaky expression that may be coming directly from the T7 promoter through directed evolution efforts may prove beneficial. Addressing this would allow the use of higher strength RBS sequences without concern for leak, thereby enabling improved yields. While the expression of toxic or highly burdensome products has long been of interest in bioproduction and synthetic biology, and effective strategies have been implemented to accomplish this[39], to our knowledge all existing strategies only work for single-use batch culture inductions. The critical difference with our strategy of terminal differentiation is that progenitor cells continuously differentiate to replenish the population of cells expressing the toxic function, thereby allowing a toxic product to be produced continuously.

We envision this strategy to be readily applied to the expression of burdensome and toxic proteins or metabolic pathways with little to no modification of the system, simply by transformation of plasmid encoding the desired T7 RNAP-driven function. With the performance of the current redundant terminal differentiation architecture, expression of the GOI can continue for 10–16+ plate generations (~55–88+ doublings) depending on differentiation rate. This equates to the number of doublings occurring in ~100–150+ h of continuous culture with a dilution rate of $0.4 \, h^{-1}$, suggesting this could enable continuous bioproduction. Furthermore, in contrast to naive expression where cell growth must be considered in optimizing expression, with terminal differentiation this optimization can be done with production yield as the sole factor, potentially enabling higher per cell production rates. This feature naturally motivates the application of metabolic engineering strategies. While tools like flux balance analysis can inform genetic modification of strains to improve the yield of valuable chemicals[40], these strategies naturally must be concerned with the growth of the organism. However, with terminal differentiation, genomic and metabolic knobs could be tuned to maximize yield without regard for the long-term viability of the cells. CRISPR/Cas systems have been demonstrated to allow activation and repression in *E. coli*, have been applied in metabolic engineering efforts, and could be co-opted in this context[41,42]. We are excited both by these engineering opportunities that are enabled by the ability to neglect cell growth in the producer cell population, and for future development of terminal differentiation to further extend the evolutionary stability of engineered functions in a general manner.

## Methods

### Strains and constructs

The wild-type *E. coli* strain JS006 was the base strain for the construction of all differentiation and naive circuit strains[43]. Constructs were assembled with a combination of Golden Gate and Gibson assembly using 3G assembly[44], and were integrated into the *E. coli* genome using clonetegration[45]. Because the R6K origin used for propagation of pOSIP plasmids from the clonetegration method of genomic integration is the same origin in our differentiation architecture R6K plasmid, we PCR amplified pOSIP backbones in two pieces (RW.posX.FL.F/RW.pos.rmR6K.R and RW.pos.rmR6K.F/RW.pos1.FL.R for pOSIP CT and KH; RW.posX.FL.F/RW.O.s2.R and RW.O.s1.F/RW.pos1.FL.R for pOSIP KO; Supplementary Data 3), removing the R6K origin, for use in Gibson assemblies with desired inserts. For Gibson assembly with linear pOSIP pieces, POS1 and POSX were used as terminal adapters instead of UNS1 and UNSX. The 1x naive and 1x differentiation strains were constructed by integration at the P21 (T) landing site, and the 2x naive and 2x split-π protein differentiation

strains by additional integration at the HK022 (H) landing site. 1x differentiation and 2x differentiation strains were integrated two additional times with the inducible Bxb1-LAA expression construct at the primary and secondary phage 186 (O) landing sites (Supplementary Fig. 5, Supplementary Table 1). Following transformation, integrations were checked via colony PCR with the pOSIP p4 primary corresponding to the landing site and a reverse primer common to all pOSIP plasmids (RW.pOSIPchk.rev)[45]. Fidelity of integrations was checked with a combination of sequencing and functional screening prior to transformation with pE-FLP to excise the antibiotic resistance cassette and integration module, and integration of subsequent constructs. Final strains (eRWnaive1X, eRWnaive2X, eRWdiff1X, eRWdiff2X) were whole-genome sequenced with MinION using the Rapid Barcoding Kit (Nanopore SQK-RBK004) for verification. Reads were assembled with Flye version 2.8.3 (https://github.com/fenderglass/Flye/) and mapped to reference genomes containing intended genomic insertions in Geneious Prime 2021.1.

Modified MoClo[46] compatible parts for T7 RNAP, integrase attachment sites, and terminators were generated with standard molecular biology techniques (PCR, Gibson, oligo annealing, and phosphorylation), and modified UNS adapters used for the construction of polycistronic or inverted transcriptional units. The R6K-CmR backbone was constructed with Golden Gate using an R6K origin amplified from the pOSIP plasmids. Sequences for Bxb1 integrase attachment sites attB and attP were obtained from Ghosh[47]. NahR$^{AM}$, LasR $^{AM,}$ and LacI $^{AM,}$, and their corresponding evolved promoters P$_{SalTTC}$, P$_{LasAM}$, and P$_{Tac}$ were provided by Adam Meyer[28]. The CIDAR MoClo Parts Kit, which includes various promoter, RBS, CDS, and terminator parts used in the constructs described, were provided by Douglas Densmore (Addgene kit 1000000059). A summary of plasmids used in this study is shown in Supplementary Table 3, and primer sequences are provided in Supplementary Data 3.

**Differentiation experiments**
Chemically competent cells were prepared from the naive and differentiation strains grown in LB without selection, with differentiation strains induced with 30 nM Las-AHL to allow π-protein expression for R6K plasmid replication. 1x and 2x naive strains were transformed with ColE1-KanR-P$_{T7}$-GFP or ColE1-AmpR-P$_{T7}$-GFP and plated on LB with 50 μg/mL kanamycin or 100 μg/mL carbenicillin, respectively. Differentiation strains were co-transformed with R6K-CmR-mScarletI and ColE1-KanR-P$_{T7}$-GFP or ColE1-AmpR-P$_{T7}$-GFP, recovered in SOC with 30 nM Las-AHL, and plated on LB agar with 34 μg/mL chloramphenicol, 30 nM Las-AHL (3OC12-HSL, Sigma O9139), and 50 μg/mL kanamycin or 100 μg/mL carbenicillin, respectively. Eight independent colonies were picked from each transformation and grown in 300 μL LB in 96 square deep well plates (Southern Labware SKU# 502062) sealed with breathable film (Diversified Biotech BERM-2000) for 8 h at 37 °C. Naive strains were grown in LB with the appropriate antibiotic, and differentiation strains were grown in LB with chlor and carb or kan with 10 nM Las-AHL (see Supplementary Method 1). Following outgrowth, cells were diluted 1:50 into experimental conditions with varying concentrations of IPTG (Gold I2481C) and salicylate (Sigma S3007). Cells were diluted every 8 h for sixteen total growths in constant antibiotic and induction conditions, and sfGFP (485/515 nm), mScarlet (565/595 nm), and OD700 measured by taking 50 μL aliquots of endpoint culture and measuring in 384 well black wall clear bottom microplates (Thermo Scientific 142761) on a Biotek Synergy H1 plate reader using Gen5 software version 3.10.06. An average of two reads for each measurement in each well was used. 1X LB media (ThermoFisher 12795084) sterilized by autoclaving was used for all experiments.

**dnaseI expression and quantification**
Chemically competent 1x and 2x naive strains, and 1x and 2x differentiation strains were transformed with 10 ng of Cole1-AmpR-P$_{T7}$-GFP

or 10 ng Cole1-AmpR with insulated P$_{T7}$ dnaseI, and all or 10 percent plated on LB carb. Plates with more than 1000 colonies on the 10 percent plate were reported as >10$^4$ cfu. For dnaseI expression experiments, 2x split-π differentiation cells were co-transformed with an empty R6K-CmR plasmid and the insulated ColE1-AmpR P$_{T7}$ dnaseI expression plasmid, recovered in SOC with 30 nM Las-AHL, and plated on LB agar with carb/chlor/30 nM Las-AHL. Three independent colonies were inoculated into 3 mL LB cultures with carb/chlor/10 nM Las-AHL, outgrown for 8 h at 37 °C, and diluted 1:50 into 25 mL media with or without 20 μM salicylate and 10 μM IPTG to induce differentiation and T7 RNAP expression. After 8 h of growth, cultures were diluted 1:50 into the same conditions, and the remaining culture harvested. Wet weight of pellets after washing with PBS was recorded before storing at −20C. JS006 parental strain without the dnaseI expression plasmid was grown similarly in LB without antibiotics and inducers for negative control. The second growth of uninduced cultures was harvested after 8 h as before, and induced cultures allowed to grow for an additional 8 h as minimal growth was observed after the initial growth.

Pellets were lysed via sonication of a 33 percent (w/v) cell suspension in 10 mM Tris pH 7.5/2 mM CaCl2 with protease inhibitor (Roche, 11836170001), cleared with centrifugation at 4 C, and supernatant collected and kept on ice before assaying dnaseI activity. Buffers used for assay were as described in Kunitz[48], though to allow simultaneous measurement of many samples and to avoid problems we observed with background absorbance in crude cell lysate when performing the Kunitz assay, we developed a fluorescence-based assay similar to Vogel and Frantz[49]. Briefly, dnaseI assay buffer was prepared by diluting SYBR Safe (Invitrogen, S33102) 1:1000 into a solution of 100 mM sodium acetate/5 mM magnesium sulfate with 26.3 μg/mL calf thymus DNA (Sigma D1501). Assay plate was prepared by aliquoting 190 μL dnaseI assay buffer into 96 round-well clear bottom plates and equilibrating in the dark at 25 °C. Standards were prepared by adding various amounts of dnaseI (Invitrogen AM2222) to JS006 lysate diluted 1:10 in 0.85 percent NaCl. Samples to assay were diluted 1:10 or 1:50 in 0.85 percent NaCl, and 10 μL of sample or standard pipetted with a multi-channel pipette into triplicate wells immediately before assay. The final amount of DNA per well was 5 μg. After shaking briefly fluorescence (487/528 nM) was measured every minute for 2 h at 25 °C on a Biotek Synergy H1 plate reader using Gen5 software version 3.10.06. Fluorescence fold-change over the course of the two-hour assay was used in fitting a standard curve (Supplementary Fig. 30), and dnaseI activity calculated from appropriate dilutions.

**Model simulations**
Simulations were run in Python using systems of ODEs using Euler's method with a time step of 0.01 h. This custom ODE solver was used to allow choice of modeling mutations deterministically using first-order rate constants for mutations, or stochastically by drawing from a binomial distribution. Production was modeled as being proportional to the ratio of specific growth rate (actual growth rate after accounting for effect due to carrying capacity) to maximum growth rate for the specific cell type, and production rate was 1 arb. unit per cell per hour for all simulations regardless of burden. For terminal differentiation, the number of divisions allowed ($n$) is explicitly modeled. Immediately after differentiation, a cell has divided $i = 0$ times. Division of a cell for which $i = 0$ generates two cells with $i = 1$. Cells for which $i = n$, instead of dividing, then die at the same rate. Simulations in Fig. 1 and Supplementary Fig. 1 were modeled deterministically with constant dilution and carrying capacity limited growth for 1000 h of simulated time. For Fig. 3 and Supplementary Fig. 17–26, simulations were of batch growths with 1:50 dilutions every 8 h for 20 total growths. Growth, differentiation, and production were modeled deterministically, while all mutations were modeled stochastically by drawing from a binomial distribution. 8 replicate simulations were run for each condition. Naive 2x indicates that a cell with two functional cassettes (subscript *PP*) has

the production rate ($\beta$) and growth rate ($\mu$) of a producer cell as indicated by the burden level being modeled ($\beta_{PP} = 1; \mu_{PP} = \mu_P$). The growth and production rates of a cell with one functional and one non-functional cassette (subscript *NP*) for naive 2x are given as $\mu_{NP} = \mu_N \sqrt{\mu_P/\mu_N}$ (**1**) and $\beta_{NP} = \frac{\mu_N - \mu_{NP}}{\mu_N - \mu_P}$ (**2**), respectively.

Naive 2x* indicates that a cell with one functional cassette has the production rate and growth rate of a producer cell ($\mu_{NP} = \mu_P; \beta_{NP} = 1$). The growth and production rates of a cell with two functional cassettes are given as $\mu_{PP} = \frac{\mu_P^2}{\mu_N}$ (**3**) and $\beta_{PP} = \frac{\mu_N - \mu_{PP}}{\mu_N - \mu_P}$ (**4**), respectively. For full description of model implementation, see Supplementary Note 1.

### Characterization of circuit failure with selective plating and sequencing

Glycerol stocks from plate generation 16 were saved in 384 well plates in 20% glycerol w/v (1:3 dilution of culture with 30% w/v glycerol) and stored at −80 °C. Glycerol stocks were thawed, and 10-fold serial dilutions of three independent experimental replicates for each strain/experimental condition being assessed were prepared and plated. For 1x and 2x naive, dilutions from replicates from the high burden condition (LB Kan/50 µM IPTG) were plated on LB + kanamycin. For 1x and 2x differentiation and terminal differentiation, dilutions from replicates from the high differentiation rate/high burden condition (LB Kan/10 nM Las/50 µM IPTG/30 µM salicylate, +/- chloramphenicol) were plated on LB + Kan/10 nM Las/50 µM salicylate, + Kan/Chlor/10 nM Las, and + Kan/Chlor/10 nM Las/50 µM salicylate. Plates were imaged on a ChemiDoc MP imager for GFP fluorescence (Alexa 488 channel, 532/28 Filter, 0.04 s exposure), mScarlet fluorescence (Alexa 546 channel, 602/50 Filter, 0.3 s exposure), and for visualization of all colonies (UV Trans Illumination, 590/110 Filter. 0.488 s exposure). Multichannel RGB images were created in Image Lab (version 6.1) with equivalently transformed images (Alexa 488: high 30000, low 0; Alexa 546: high 30000, low 0; UV Trans: high 60000, low 30000, inverted), and colonies manually counted using an application (COUNT THINGS) on an iPad pro.

For each strain/condition, 6 colonies were analyzed with nanopore sequencing. Colony PCR using barcoded primers (Supplementary Data 3) was performed for all regions of interest with PrimeSTAR HS DNA Polymerase (1 min 98 °C; 30 cycles of 10 s 98 °C, 5 s 55 °C or 60 °C, 1 min/kb 72 °C; 5 min 72 °C final extension; 20 µL rxns). Amplified samples were assessed with electrophoresis (5 µL), and equivalently sized products pooled and gel extracted. Equimolar amounts per amplicon of purified products were pooled and prepared for Nanopore sequencing (Flongle flow cell on MinION) with LSK110 according to the manufacturers protocol, and super accuracy basecalling was used in MinKNOW. The portion of the ColE1 plasmid encoding $P_{T7}$ GFP and the P21 (T) integration locus were amplified for all samples; the HK022 (H) locus was amplified for 2x naive and 2x differentiation/terminal differentiation; and the primary and secondary phage 186 (O) sites were amplified for all differentiation/terminal differentiation strains. Sequences were demultiplexed and analyzed using Maple (https://github.com/gordonrix/maple) using expected reference sequences (WT amplicons for ColE1, naive T and H integrations, and primary and secondary O integrations; and WT, and recombined (correctly excised, as well as inverted) sequences for T and H differentiation integrations. As this pipeline failed for a subset of differentiation cassettes with unexpected deletion/recombination mutations, basecalled reads were annotated with barcode primer sequences and extracted with in silico PCR in Geneious (Geneious Prime 2022.1), and exported as fastq.gz files for de novo assembly using Flye 2.9 (https://github.com/fenderglass/Flye). This de novo assembly was used to analyze all T and H locus amplicons for differentiation/terminal differentiation by alignment to WT, excised, and inverted reference sequences.

### Reporting summary

Further information on research design is available in the Nature Research Reporting Summary linked to this article.

## Data availability

The sequences of all plasmids have been deposited in Genbank: pRW01 (OP654158), pRW02 (OP654159), pRW03 (OP654160), pRW04 (OP654161), pRW05 (OP654162), pRW06 (OP654163), pRW07 (OP654164), pRW08 (OP654165), pRW09 (OP654166), pRW10 (OP654167), pRW11 (OP654168), pRW12 (OP654169), and pRW13 (OP654170). The de novo assembled genomic sequences of strains eRWnaive1X (SAMN31276766), eRWnaive2X (SAMN31276767), eRW-diff1X (SAMN31276768), and eRWdiff2X (SAMN31276769) are available on NCBI, and the sequences of each genomic integration is available in Supplementary Data 1. The sequencing data described in the text and Supplementary Fig. 16 is summarized in Supplementary Data 2. Primer sequences are available in Supplementary Data 3. Source data are provided with this paper.

## Code availability

All code for running, analyzing, and plotting simulations are available on GitHub[50] [https://github.com/rlwillia/terminal-differentiation-for-evolutionary-stability] or Zenodo [https://doi.org/10.5281/zenodo.7213995].

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

## Acknowledgements

The authors would like to thank the Sim and Liu Labs at the University of California Irvine for the use of lab space and equipment; Andrey Shur and Gordon Rix for help with MinION sequencing; Justin Bois, Andy Halleran, Anandh Swaminathan, and Andrey Shur for productive conversations; and Prof. Chang Liu, John Marken, and Gordon Rix for providing comments on the manuscript. NahR$^{AM}$, LasR$^{AM}$ and LacI$^{AM}$, and their corresponding evolved promoters P$_{SalTTC}$, P$_{LasAM}$, and P$_{Tac}$ were provided by Adam Meyer[28]. The CIDAR MoClo Parts Kit, which includes various promoter, RBS, CDS, and terminator parts used in the constructs described, was provided by Douglas Densmore (Addgene kit 1000000059). This research was supported by the Army Research Office (ARO) through grants W911NF-19-2-0026 and W911NF-09-D-0001.

## Author contributions

R.W. and R.M. conceived the presented idea; R.W. cloned all strains and plasmids, planned and carried out all experiments, developed the mathematical models, ran the simulations, analyzed the data, and wrote the manuscript with input and guidance from R.M.

## Competing interests

The authors declare no competing interests.
