## [Peer Review File · Nature Communications]

Integrase-mediated differentiation circuits improve evolutionary stability of burdensome and toxic functions in *E. coli*Reviewers' Comments:

Reviewer #1:

Remarks to the Author:

This is an absolutely beautiful manuscript that describes the creation of genetic circuits that cause a fraction of a growing *E. coli* population to differentiate into either growing or nongrowing cells that can activate expression of a burdensome gene. This system belongs right alongside the "classic" genetic circuits of synthetic biology (e.g., the repressilator, toggle switches, etc.). A number of clever designs and interesting genetic parts were needed to implement the differentiation system, including split T7 RNAP and pi proteins. The components and subsystems were thoroughly tested. The system was modeled from multiple perspectives to inform its design and address further questions about plasmid stability. Overall, this work goes above and beyond in implementing and testing an interesting solution for stabilizing toxic/burdensome engineered functions. I have only minor questions and suggestions.

Specific Comments

1. The authors decide to call mutants with broken production "cheaters" and mutants with broken differentiation "defectors". These terms may be freighted with too much meaning. They make one think of social behavior and the prisoner's dilemma when there are no such interactions between cells in this system. I do not strongly object to using these terms, but it seems simpler and more in line with the evolutionary failure literature to call these mutants "nonproducers" and "nondifferentiators".
2. Regarding the Fig. 2 results, is a subtle difference between the 2x differentiation circuit and the 2x naive setup that initially only one of the recombination events occurs so there is just 1x of T7 RNAP in the 2x differentiation setup that is driving GFP expression? Later the same differentiated cell could activate the second copy, even if it has become a nonproducer in the meantime?
3. Do we know what fraction of the population of *E. coli* cells are differentiated at the end of a growth cycle at each of the differentiation rates shown in Fig. 2? Does this vary between the 1x and 2x differentiation circuits? At the highest differentiation rate, how much does "loss" of the differentiated cells slow growth of the population?
4. I think the origin of replication SBOL symbol labeled "Gen." in many figures means genome? Since plasmids are part of a cell's "genome" by most common definitions, "chromosome" would be more precise. There is a specific SBOL way of curving the lines outside of the synthetic construct to indicate "in the chromosome" versus "in a plasmid". It might be helpful for some "circuit diagram" figures in the future. Things are clear enough for this paper if "Gen." is explained in a main text caption.
5. Expression of mutation rates in the model and related figures as per hour versus per cell division is different from normal conventions in microbiology.
6. Fig. S3 seems to be missing descriptions of panels D and E.
7. There are some unformatted \cite tags in the modeling section of the supplement.
8. The authors should consider depositing their simulation code as a supplement to the paper or (even better) put it into a code/data repository.

Reviewer #2:

Remarks to the Author:

The authors constructed a genetic circuit that "differentiates" *E. coli* to a new physiological state using an integrase-mediated recombination system. The authors aim to improve the evolutionary stability of

high-burden gene expression circuits in microbial populations by dividing growth and production between different subpopulations. More specifically, the “progenitor” cells focus on growing and maintaining the population size, while the “terminally differentiated” producer cells direct their metabolic resources solely into production. Moreover, because the producer cells do not grow, the escape mutants cannot grow in number, even if the burdened producer cells mutate. The central design strategy is clever.

To implement this concept experimentally, the authors used bacteriophage serine integrases (Bxb1) to “differentiate” the engineered *E. coli* from the original progenitor state into the producer mode. All progenitor cells carry 4 components: (1) Bxb1 encoded under salicylate promoter; (2) ColE1 plasmids encoding the gene of interest (GOI) under T7 promoter; (3) *tac* promoter driving the expression of T7 RNAP, which is interrupted by split inteins carrying *pir* (responsible for R6K plasmid replication); and (4) R6K plasmid encoding antibiotic resistance gene. Upon a differentiation signal (i.e., salicylate), Bxb1 irreversibly excises out *pir* so that T7 RNAP recovers function, which then initiates the GOI expression. At the same time, the *pir* removal inhibits R6K plasmid replication so that the cells can no longer survive in the presence of antibiotics. The notion of simultaneous GOI activation and growth suppression is the key feature of the terminal differentiation that can improve the evolutionary stability to plasmid-based circuits.

The authors also implemented a few other safeguards for improving genetic stability. To reduce the chance of a complete loss-of-function via escape mutation, the authors duplicated the circuit cassette into the host genome. The integrase and T7 RNAP-*pir* cassettes were genomically integrated to stabilize their expression level and reduce the probability of loss through segregation error. To ensure that R6K plasmid-free subpopulation does not grow out and overtake the population due to growth advantage, the authors added appropriate antibiotics to suppress the mutant growth.

The authors conducted an extensive mathematical analysis on population dynamics and their proposed technology’s production yield against the naïve counterpart, which express GOI under a simple promoter. Then, they tested the evolutionary stability of their circuit in a set of long-term experiments to validate that their terminal differentiation architecture suppressed the emergence of cheaters. The authors claim that their architecture confers evolutionary stability against other sources of failures, such as plasmid loss due to communal antibiotic degradation and plasmid mutation.

Because evolutionary stability is a major issue when engineering microbial community for producing toxic chemicals to host, the authors propose that their design principle can improve toxin production in *E. coli* by significantly reducing the leaky expression of the toxin gene. The authors demonstrated that their architect improved the production level of their target toxin *dnaseI* by ~ 1000 -fold.

The design principle that the authors are introducing is intriguing. To the extent I can understand, their demonstrations are overall convincing. However, the authors should improve the presentation of their work to deliver their major points more coherently and concisely.

Comments:

1. The authors should substantially revise (mostly simplify) the manuscript for better readability. I had a really difficult time reading the paper. It’s a pity as I feel the central design concept is actually simple (in a good way) but the simple message is lost in all the data that are presented.

Several sections of the manuscript are difficult to read because sentences are generally too long, contain too much information, use ambiguous expressions, or lack appropriate punctuations. A few sentences carry typos. The authors also use certain expressions too frequently (e.g., too many uses of “indefinite”, “no opportunity”, “uniquely”, etc.).

2. In the last paragraph of the introduction, it may be helpful to use some graphical illustrations to explain the nature-inspired design concept more clearly (how differentiation helps with improving evolutionary stability). Also, it would be nice to give some specific examples of how "the use of differentiation to coordinate such division of labor is a recurring strategy used by microorganisms" as stated in the 2nd paragraph of the introduction. The authors emphasize borrowing the design principle from nature but do not provide sufficient justification on how the design principle improves evolutionary stability.

3. It will be helpful to introduce some of the technical terminologies in the main text. For instance, what's the difference between differentiation vs. terminal differentiation? These terminologies deliver central concepts to the proposed technology, but the authors provide insufficient information.

4. I have major reservations about using "cheater" and "defector" to refer to two different types of cheating. In the evolutionary biology literature, these two terms are often used interchangeably. The usage of these terms in the paper makes the paper harder to understand.

5. I think that the first figure needs to be rearranged to deliver the central message in a more concise and effective manner.

First, the text mentions that the "terminal differentiation" model inhibits the cheaters' growth. However, Fig 1C crosses out (in red) only one of the two of the cheater's growth. I am not sure if this was intended because it is not specified in the text.

Second, in the 4th paragraph of the first results section, the authors write "the duration of function for terminal differentiation, strikingly, is unaffected by burden, as cheaters are never able to expand in the population (Figure 1F,I,L; Figure S1D)." What is "the duration of function"? Also, because the colors for cheaters and defectors in Fig1 are so similar that it took me a long time to understand what the authors try to point out (Fig 1F and 1I show flat, near-zero light green curves but it's so hard to see that, especially with the very eye-catching dark green curves).

Third, unless $k_{MD} = 0$, the production rate of the authors' proposed model (terminal differentiation; Fig 1C,F,I,L) is lower than the regular differentiation model (Fig 1B,E,H,K). This seems to be a major caveat of the proposed strategy, as the "defectors" take over more quickly and significantly in the absence of the "cheaters". Especially since the authors state that "While eliminating this mutation may not be possible," I believe that Fig 1 needs to be improved to highlight the significance and importance of their proposed system.

6. When first introducing "bacteriophage serine integrases" in the results section, it would be nice to indicate that it's called "Bxb1" since the figure uses this abbreviation. It may also be helpful to use annotations from the figure in the text to help guide the readers.

7. The authors should take more time to explain the basic functionality of their platform before jumping into the more complicated, long-term data in the main text.

8. In Figure 2, the authors aim to demonstrate the evolutionary stability of their circuits compared to the naïve and non-terminal differentiation counterparts. However, Fig 2 does not convey this message very effectively. For instance, the authors use the cumulative (total) sfGFP as their readout for evolutionary stability, but the naïve version circuit outperforms their proposed circuit at a lower burden based on this metric. Instead, it may be helpful to show the change in sfGFP intensity (or at least highlight the notion of change in production rate), because the authors can reason that the slowdown of sfGFP production may signal when the system has broken to function properly.

9. The authors argue that the reduced productivity (Fig 2,3) is attributed to broken circuits. The sequencing information on the broken vs. intact circuits at the end of the long term would be compelling to support their argument, especially because the authors computationally model plasmid mutation as a major source of circuit instability in Fig 3. Their experimental data (Fig 3F) does not

encompass the sequence-level validation of circuit stability. The authors should also examine the fraction of intact vs. cheater populations to validate the evolutionary stability of their engineered population.

Such validation is crucial the authors add chloramphenicol (cm) to select for the R6K plasmid during the entire course of their long-term experiments, whereas the differentiation eliminates the R6K-free population. These opposing two forces can cause severe stress on the cells that they are more prone to break.

Moreover, in Figure 3, the authors specifically investigate on the effect of plasmid mutation using their mathematical model, but in Figure 2, it is ambiguous if they only consider the genome-integrated components when they refer to the evolutionary stability of the "circuit". Without explicitly examining where in the circuit was compromised, it is inconclusive to claim what were the contributing factors to the observation described in Fig 2 and 3.

10. The lower burden naïve population seems to do better or just as good in GFP production compared to their proposed architecture unless the final product is especially detrimental to the host. Perhaps the authors should focus and elaborate more on the content shown in Figure 4 to better demonstrate their system's significance.

11. Figure 4A should be plotted in bar graph (y axis representing CFU). The two axes should be flipped in Figure 4B to match a uniform style throughout the text.

[1] Summary of Revisions

• Main Figures

- 1) Figure 1 has been updated to more clearly and effectively communicate the main messages
 - a. Cartoon depictions have been updated with clearer color schemes, cells formerly labeled as cheaters are now non-producers, cells formerly labeled as defectors are now labeled as non-differentiators, progenitors that have incurred a burden mutation distinguished as “progenitors*”
 - b. Panels D-F color scheme has been changed to match panels A-C. Panels G and H now illustrate now show the total production and total duration of function, respectively, for the three architectures as a function of burden. These panels now clearly highlight the robustness of terminal differentiation to burden, and the benefit of reducing the differentiation mutation rate for this architecture on evolutionary stability.
- 2) Figure 2 panels A-C now plot the endpoint GFP production at each plate generation rather than cumulative production, and the color scheme updated for improved clarity.
- 3) Figure 3C-E have had minor layout and color changes. Panel F color scheme has been changed to match Figure 2.
- 4) Figure 4B plot is now oriented with dnaseI activity on the y-axis.
- 5) Genomic integrations are now depicted with the SBOL curved symbol.

• Manuscript

- 6) Throughout the manuscript, we now use the term *non-producers* instead of *cheaters* to refer to the naïve population that has incurred burden mutations and the differentiated population that has incurred burden mutations. We also use *non-differentiators* in place of *defectors* to refer to progenitor cells that have incurred differentiation mutations and no longer differentiate.
- 7) The introduction now includes a discussion of natural examples of division of labor through differentiation, and the important distinctions between natural examples and our engineered strategy.
- 8) The introduction and results section have been updated to include a more thorough explanation of the functionality of our engineered circuits, the motivation behind the terminal differentiation strategy, and the distinction between differentiation and terminal differentiation. This includes discussion of a new experiment (Figure S28) in which 1x/2x differentiation/terminal differentiation was characterized using a Tecan plate reader to obtain time-course growth and fluorescence data (to determine impact of differentiation rate on growth rate), and flow cytometry to characterize the fraction of differentiated cells generated at varying induction levels of Bxb1 integrase.
- 9) The results section now includes a discussion of selective plating assays performed on glycerol stocks saved after plate generation 16 from the Figure 2 experiment (to enumerate producer/non-

producer populations for naïve circuits, and progenitor/producer, non-producer, and non-differentiator populations for differentiation and terminal differentiation circuits), and sequencing performed on colonies to characterize mutation mechanisms. Full results of this experiment are described in Figure S29 and a supplementary excel file. The experimental methods for this are described in the Methods section under *Characterization of circuit failure with selective plating and sequencing*.

- 10) Mutation rates (k_{M_B} , k_{M_D} , k_{M_I}) have now been corrected to reflect the dependence on growth rate. For example $k_{M_B} = 10^{-6} \times \mu = 10^{-6} \times \ln(2)/T_d$, where μ is the growth rate which varies over time due to the effect of carrying capacity, and T_d is the doubling time.

- **Supporting Information**

- 11) Figure S5 has been updated to improve clarity (panel A uses SD shading rather than plotting individual replicates), and panel C added to show dose response between IPTG induction and GFP production for 1x and 2x naïve circuits.
- 12) Figure S11 has been updated to match Figure 2
- 13) Figure S28 was added describing the plate reader/flow-cytometry experiment in (8) above. Supplementary methods for this experiment are included in the main supporting information file under *Characterization of Bxb1 induction dose-response on differentiated cell fraction*.
- 14) Figure 29 was added describing the selective plating and sequencing analysis performed on glycerol stocks saved after the conclusion of the Figure 2 experiment (described above in (9)). Complete data and sequencing results are described in the included supplementary excel file.
- 15) All raw data and code for data analysis and plotting has been included as a zip file. Additionally code for simulations will be deposited on GitHub.

[2] Response to *Reviewers' Comments*

For Reviewer 1:

This is an absolutely beautiful manuscript that describes the creation of genetic circuits that cause a fraction of a growing E. coli population to differentiate into either growing or nongrowing cells that can activate expression of a burdensome gene. This system belongs right alongside the “classic” genetic circuits of synthetic biology (e.g., the

repressilator, toggle switches, etc.). A number of clever designs and interesting genetic parts were needed to implement the differentiation system, including split T7 RNAP and pi proteins. The components and subsystems were thoroughly tested. The system was modeled from multiple perspectives to inform its design and address further questions about plasmid stability. Overall, this work goes above and beyond in implementing and testing an interesting solution for stabilizing toxic/burdensome engineered functions. I have only minor questions and suggestions.

► We appreciate these generous comments and summary.

1) *The authors decide to call mutants with broken production “cheaters” and mutants with broken differentiation “defectors”. These terms may be freighted with too much meaning. They make one think of social behavior and the prisoner’s dilemma when there are no such interactions between cells in this system. I do not strongly object to using these terms, but it seems simpler and more in line with the evolutionary failure literature to call these mutants “nonproducers” and “nondifferentiators”.*

► We appreciate the critique, and have changed the language in the text and figures to use “non-producers” and “non-differentiators” as you have suggested.

2) *Regarding the Fig. 2 results, is a subtle difference between the 2x differentiation circuit and the 2x naïve setup that initially only one of the recombination events occurs so there is just 1x of T7 RNAP in the 2x differentiation setup that is driving GFP expression? Later the same differentiated cell could activate the second copy, even if it has become a nonproducer in the meantime?*

► Yes that is correct, and this phenomenon is incorporated into our modeling. While this is not expected to occur with 2x terminal differentiation due to the limitation of growth after a single copy of T7 RNAP is activated, this could occur as you state with 2x differentiation. In our modeling, if one cassette is activated for 2x differentiation, the production rate and growth rate are equivalent to 1x naïve, and if both cassettes are activated, the production/growth rates are equivalent to naïve 2x*, as described in the Figure 3 caption and supplementary information.

3) *Do we know what fraction of the population of E. coli cells are differentiated at the end of a growth cycle at each of the differentiation rates shown in Fig. 2? Does this vary between the 1x and 2x differentiation circuits? At the highest differentiation rate, how much does “loss” of the differentiated cells slow growth of the population?*

► These are great questions. Because of the scale and timing of the experiments described in Figure 2 and Figure S11, it was not feasible to do flow cytometry or collect growth curves to access this information during the experiments. To address this question, we performed a smaller scale experiment with similar conditions. In this experiment (Figure S28), we selected 4 salicylate concentrations (0, 10, 20, 30 μ M), 2 IPTG concentrations (10, 50 μ M), and the KanR version of the ColE1 plasmid, and grew the cells for 3 passages in a Tecan plate reader to allow real-time monitoring of growth. From the plate reader data, we determined growth rate, terminal OD, and terminal GFP production, and we further analyzed the population composition with flow cytometry to assess the fraction of differentiated producer cells.

Figure S28: Characterization of differentiation rates and the impact of differentiation on population growth rates for 1x and 2x differentiation architectures. **(A)** 1x differentiation (left) and 2x differentiation (right) strains were co-transformed with ColE1-KanR-P_{T7} GFP and R6K-CmR-mScarlet, and plated on LB +kan/chlor/30 nM Las-AHL. **(B-E)** Colonies were outgrown in LB +kan/chlor/10 nM Las-AHL before 50x dilution into experimental conditions in LB kan with chlor (terminal differentiation) or without chlor. Cells were grown in 96 well black-wall clear bottom matriplates (Brooks

MGB096-1-2-LG-L) in 300 μ L LB media with varying salicylate (0, 10, 20, 30 μ M), and IPTG (10 μ M: blues, 50 μ M: oranges) in a Tecan plate reader at 37C with 6 min shaking (5 min linear, 1 min orbital) between reads (OD700, sfGFP: 485nm/515nm, mScarlet: 565nm/595nm) at 10 min intervals. Cells were diluted 1:50 every 8 h for 3 plate growths, and cells diluted 1:100 in 0.9% NaCl for flow cytometry analysis immediately after each growth. Endpoint OD normalized GFP (**B**), GFP+ fraction as determined by flow cytometry (**C**), endpoint OD700 (**D**), and growth rate (**E**) as determined by fitting an exponential growth model to time-course OD700 measurements (trimmed to OD700<0.2) are plotted for each plate generation (Plate 1-3 left to right, light to dark). Mean +/- SD error bars, with individual data points plotted with small circles.

For both the 1x and 2x differentiation strains, we observed very little leaky differentiation (~1% or less GFP+ cells). We also note that the differentiation rate is very sensitive to salicylate concentration between 10 μ M and 30 μ M, and therefore small differences in concentration can result in noticeable differences in differentiation rate. In this experiment, medias containing all inducers and antibiotics were with the exception of salicylate were prepared (Kan/Las-AHL/10 μ M IPTG, Kan/Las-AHL/50 μ M IPTG, Kan/Chlor/Las-AHL/10 μ M IPTG, Kan/Chlor/Las-AHL/50 μ M IPTG), and 50x stock solutions of each salicylate concentration prepared in each media. 6 μ L of inducers were pipetted with a multichannel, followed by media, then 6 μ L cells (1:50 dilution) for a total of 300 μ L. Consequently while there may be small differences in the concentration of salicylate across media conditions that influence experimental results, 1x differentiation and 2x differentiation should be directly comparable for each experimental condition. Additionally, though exact comparison to the data in Figure 2 is not possible due to (1) differences in media preparation, including separately prepared batches of LB media, and stocks/dilutions of inducers/antibiotics (salicylate, IPTG, Las-AHL, kanamycin, chloramphenicol), and (2) differences in growth conditions, with cells in Figure 2 grown in 96-well deep square-well plates (Southern Labware 502062) sealed with breathable film in a shaking incubator, and cells in the experiment above grown in 96-well square-well glass bottom plates (Brooks MGB096-1-2-LG-L) in a Tecan plate reader, general trends are still informative. After the first plate growth, 1x differentiation without chloramphenicol has ~0.2%, ~18%, ~64%, and ~94% differentiated producer cells (GFP+) with 0, 10, 20, 30 μ M salicylate in the lower burden condition (10 μ M IPTG), with similar though slightly lower percentages in the higher burden condition (~0.2%, ~13%, ~56%, and ~90%). 2x differentiation is somewhat more sensitive to salicylate, with ~1%, ~30%, ~76%, and ~97% in the lower burden case, and ~1%, ~43%, ~74% and ~97% in the higher burden condition. For 1x terminal differentiation (with chloramphenicol), these percentages are somewhat higher than non-terminal differentiation, with ~0.3%, ~29%, ~74%, and ~98% for the lower burden case, and ~0.2%, ~25%, ~68%, and ~95% for the higher burden case. For 2x terminal differentiation, GFP+ percentages were more similar to the higher burden than lower burden non-terminal differentiation case, with ~0.4%, ~44%, ~83%, and ~98% in the lower burden case, and ~0.3%, ~39%, ~75%, and ~96% in the higher burden case.

To determine the impact of differentiation rate on growth rate, we fit background subtracted OD700 growth curves truncated to OD700<0.2 with an exponential growth model. As differentiation has not been induced prior to the first plate generation, we will discuss the growth rates observed from the second plate generation where the initial growth rate reflects the distribution of progenitor and differentiated cells achieved at each condition. For 1x differentiation, growth rates were ~1.28, ~1.35, ~1.31, and ~1.22 h⁻¹ in the lower burden case with 0, 10, 20, and

30 μ M salicylate respectively, and ~ 1.26 , ~ 1.28 , ~ 1.24 and ~ 1.02 h⁻¹ in the higher burden case. For 2x differentiation, growth rates were ~ 1.25 , ~ 1.28 , ~ 1.25 , and ~ 1.2 h⁻¹ in the lower burden case, and ~ 1.20 , ~ 1.22 , ~ 1.19 , and ~ 0.97 in the higher burden case. For 1x terminal differentiation, growth rates were ~ 1.28 , ~ 1.29 , ~ 1.21 , and ~ 1.04 h⁻¹ with lower burden, and ~ 1.23 , ~ 1.22 , ~ 1.18 , ~ 0.89 h⁻¹ for the higher burden. For 2x terminal differentiation, growth rates were ~ 1.13 , ~ 1.16 , ~ 1.05 , and ~ 0.64 h⁻¹ for the lower burden, and ~ 1.15 , ~ 1.15 , ~ 1.1 , and ~ 0.52 h⁻¹ for the higher burden. Here we note that for both 1x and 2x differentiation without chloramphenicol, the growth rate is reduced to a greater extent in the higher burden condition, most notably at the highest differentiation rate (30 μ M salicylate). Relative to the corresponding case of no induction of differentiation, the highest differentiation induction reduced the growth rate by $\sim 5\%$ and $\sim 19\%$ for 1x differentiation with low and high burden, respectively; $\sim 4\%$ and $\sim 19\%$ for 2x differentiation; $\sim 19\%$ and $\sim 28\%$ for 1x terminal differentiation; and $\sim 43\%$ and $\sim 55\%$ for 2x terminal differentiation. As we would expect, the impact of chloramphenicol on growth rate is more exaggerated for the lower burden case. We further observe that while for 1x terminal differentiation with 30 μ M salicylate the growth rate rebounds in third plate generation while the percentage of GFP+ cells declines, indicating that non-differentiators have emerged, however with 2x terminal differentiation the growth rate has not yet rebounded.

We also observed in this experiment a decline in the percentage of mScarlet+ cells in both 1x and 2x differentiation with and without chloramphenicol when differentiation was not induced (0 μ M salicylate), with this being much more pronounced with 2x differentiation and terminal differentiation. This indicates that the induction level of π protein with Las-AHL in this experiment was not sufficient to reliably maintain a high copy number of the R6K plasmid with the 2x differentiation circuit. This likely explains our observation that the growth rates observed without salicylate induction for 2x differentiation were lower with chloramphenicol selection than without, while this was not observed with 1x differentiation. However, in experiments conducted to inform the choice of 10nM Las-AHL prior to the experiments described in Figure 2 and Figure S11 (performed in the exact manner as for Figure 2 and Figure S11), we did not observe a decline in mScarlet fluorescence over 3 plate generations (Figure S7-S8). This indicates that some experimental variable between these experiments (LB media batch, inducer preparation, plate growth conditions, etc.) resulted in lower levels of π -protein/R6K plasmid in the experiment described above than for those described in Figure 2, S7-S8, and S11. Though the conditions of this experiment appear to allow loss/reduced copy number of R6K plasmid to occur without inactivation of π -protein expression, we also observed evidence for this in experiments addressing comments from Reviewer #2.

Despite this caveat to the experiment, it is still informative for the approximate fraction of differentiated cells achieved at varying salicylate concentrations and the affect this has on growth rate, but perhaps more accurately so for non-terminal differentiation. Generally 2x differentiation has a somewhat higher differentiation rates than 1x differentiation for a given induction level. While this is true, in the experiments shown in Figure 2 and Figure S11 we used four concentrations which spanned the dynamic range of differentiation rates for both 1x and 2x differentiation (10, 15, 20, and 30 μ M salicylate).

4) *I think the origin of replication SBOL symbol labeled “Gen.” in many figures means genome? Since plasmids are part of a cell’s “genome” by most common definitions, “chromosome” would be more precise. There is a specific SBOL way of curving the lines outside of the synthetic construct to indicate “in the chromosome” versus “in a plasmid”. It might be helpful for some “circuit diagram” figures in the future. Things are clear enough for this paper if “Gen.” is explained in a main text caption.*

► We have updated the circuit diagrams to use the SBOL curved lines to depict genomic integrations.

5) *Expression of mutation rates in the model and related figures as per hour versus per cell division is different from normal conventions in microbiology.*

► Thank you for making this observation. In our simulations, we model cell growth with ODEs as a continuous process, and do not model discrete cell divisions. However, because we model carrying capacity limited growth, the growth rate is not constant, and we incorporate this into the mutation rate to reflect this. For example,

where X_P is a producer cell, X_N is a non-producer cell, k_{M_B} is the burden mutation rate, μ_P (h^{-1}) is the producer maximum growth rate, X_{tot} is the total population size, and K is the carrying capacity. We see here then that the parameter k_{M_B} does not have units, as μ_P is what gives the rate expression the units of h^{-1} . In the modeling figures, however, we use the mutation rate parameters (k_{M_B} , k_{M_D} , k_{M_I}) to describe the aggregate rate, as we have now clarified in the supplement. Our mutation rates, for example, should be written as $k_{M_B} = 10^{-6} \times \mu$. As growth rate and doubling time (T_d) are related by $T_d = \ln(2)/\mu$, this is equivalent to $k_{M_B} = 10^{-6} \times \ln(2)/T_d$. Because the exact mutation rates were not important for the conclusions drawn from the simulation results, we have chosen to make this clarification in the figure legends. As well, the x-axis of Figure S1 panel G has been corrected to k_{M_B}/μ , instead of $k_{M_B}(\text{h}^{-1})$.

6) *Fig. S3 seems to be missing descriptions of panels D and E.*

► We have corrected the figure caption to include these descriptions.

7) *There are some unformatted \cite tags in the modeling section of the supplement.*

► Thank you, we have corrected these.

8) *The authors should consider depositing their simulation code as a supplement to the paper or (even better) put it into a code/data repository.*

► All code for running simulations has been included in the supplementary zip file, and will also be deposited on Github.

For Reviewer 2:

The authors constructed a genetic circuit that “differentiates” E. coli to a new physiological state using an integrase-mediated recombination system. The authors aim to improve the evolutionary stability of high-burden gene expression circuits in microbial populations by dividing growth and production between different subpopulations. More specifically, the “progenitor” cells focus on growing and maintaining the population size, while the “terminally

differentiated” producer cells direct their metabolic resources solely into production. Moreover, because the producer cells do not grow, the escape mutants cannot grow in number, even if the burdened producer cells mutate. The central design strategy is clever.

To implement this concept experimentally, the authors used bacteriophage serine integrases (Bxb1) to “differentiate” the engineered E. coli from the original progenitor state into the producer mode. All progenitor cells carry 4 components: (1) Bxb1 encoded under salicylate promoter; (2) ColE1 plasmids encoding the gene of interest (GOI) under T7 promoter; (3) tac promoter driving the expression of T7 RNAP, which is interrupted by split inteins carrying pir (responsible for R6K plasmid replication); and (4) R6K plasmid encoding antibiotic resistance gene. Upon a differentiation signal (i.e., salicylate), Bxb1 irreversibly excises out pir so that T7 RNAP recovers function, which then initiates the GOI expression. At the same time, the pir removal inhibits R6K plasmid replication so that the cells can no longer survive in the presence of antibiotics. The notion of simultaneous GOI activation and growth suppression is the key feature of the terminal differentiation that can improve the evolutionary stability to plasmid-based circuits.

The authors also implemented a few other safeguards for improving genetic stability. To reduce the chance of a complete loss-of-function via escape mutation, the authors duplicated the circuit cassette into the host genome. The integrase and T7 RNAP-pir cassettes were genomically integrated to stabilize their expression level and reduce the probability of loss through segregation error. To ensure that R6K plasmid-free subpopulation does not grow out and overtake the population due to growth advantage, the authors added appropriate antibiotics to suppress the mutant growth.

The authors conducted an extensive mathematical analysis on population dynamics and their proposed technology’s production yield against the naïve counterpart, which express GOI under a simple promoter. Then, they tested the evolutionary stability of their circuit in a set of long-term experiments to validate that their terminal differentiation architecture suppressed the emergence of cheaters. The authors claim that their architecture confers evolutionary stability against other sources of failures, such as plasmid loss due to communal antibiotic degradation and plasmid mutation.

Because evolutionary stability is a major issue when engineering microbial community for producing toxic chemicals to host, the authors propose that their design principle can improve toxin production in E. coli by significantly reducing the leaky expression of the toxin gene. The authors demonstrated that their architect improved the production level of their target toxin dnaseI by ~ 1000-fold.

The design principle that the authors are introducing is intriguing. To the extent I can understand, their demonstrations are overall convincing. However, the authors should improve the presentation of their work to deliver their major points more coherently and concisely.

- We thank the reviewer for their thoughtful summary of the work. We would like to clarify that inactivation of pir expression does not immediately lead to growth cessation in the case of terminal differentiation, but only after the R6K plasmid dilutes through cell growth and division and expression of the antibiotic resistance is lost. We would also like to clarify that in Figure 4, we do not make a direct comparison between naïve and differentiation mediated expression of dnaseI. This is because the leaky expression of T7 RNAP in the naïve case prevents cells transformed with plasmid from forming colonies. The negative control we show instead is from the dnaseI assay being performed on lysate from WT cells.

- 1) *The authors should substantially revise (mostly simplify) the manuscript for better readability. I had a really difficult time reading the paper. It's a pity as I feel the central design concept is actually simple (in a good way) but the simple message is lost in all the data that are presented.*

Several sections of the manuscript are difficult to read because sentences are generally too long, contain too much information, use ambiguous expressions, or lack appropriate punctuations. A few sentences carry typos. The authors also use certain expressions too frequently (e.g., too many uses of "indefinite", "no opportunity", "uniquely", etc.).

- ▶ Thank you for the comments. It is our goal to make this manuscript as clear and understandable as possible, and we have made substantial revisions that we feel convey the main points more effectively.
- 2) *In the last paragraph of the introduction, it may be helpful to use some graphical illustrations to explain the nature-inspired design concept more clearly (how differentiation helps with improving evolutionary stability). Also, it would be nice to give some specific examples of how "the use of differentiation to coordinate such division of labor is a recurring strategy used by microorganisms" as stated in the 2nd paragraph of the introduction. The authors emphasize borrowing the design principle from nature but do not provide sufficient justification on how the design principle improves evolutionary stability.*
 - ▶ In addressing your comments regarding Figure 1 (comment #4), we feel the cartoons (Figure 1A-C) combined with the simulation results effectively convey how this strategy improves evolutionary stability of burdensome functions. We specifically highlight the manner in which naïve expression, differentiation, and terminal differentiation fail. While naïve expression fails due to the selection of burden mutations which generate non-producers and differentiation can fail both from burden mutations (generating non-producers) and differentiation mutations which generate non-differentiators, terminal differentiation **only** fails from differentiation mutations. The simulations in this figure now clearly show the benefits of terminal differentiation, (1) robustness to burden, and (2) the ability to improve the evolutionary stability of any arbitrary function by reducing the rate at which non-differentiators emerge.
 - 3) *It will be helpful to introduce some of the technical terminologies in the main text. For instance, what's the difference between differentiation vs. terminal differentiation? These terminologies deliver central concepts to the proposed technology, but the authors provide insufficient information.*
 - ▶ Thank you for the comment, we have revised the manuscript to more clearly describe the circuits described in this paper, and further emphasize the distinction between differentiation and terminal differentiation.
 - 4) *I have major reservations about using "cheater" and "defector" to refer to two different types of cheating. In the evolutionary biology literature, these two terms are often used interchangeably. The usage of these terms in the paper makes the paper harder to understand.*
 - ▶ To address this, we have changed the language in the text and figures, using "non-producer" in place of "cheater", and "non-differentiator" in place of defector.

- 5) *I think that the first figure needs to be rearranged to deliver the central message in a more concise and effective manner. First, the text mentions that the “terminal differentiation” model inhibits the cheaters’ growth. However, Fig 1C crosses out (in red) only one of the two of the cheater’s growth. I am not sure if this was intended because it is not specified in the text.*
- Second, in the 4th paragraph of the first results section, the authors write “the duration of function for terminal differentiation, strikingly, is unaffected by burden, as cheaters are never able to expand in the population (Figure 1F,I,L; Figure S1D).” What is “the duration of function”? Also, because the colors for cheaters and defectors in Fig1 are so similar that it took me a long time to understand what the authors try to point out (Fig 1F and 1I show flat, near-zero light green curves but it’s so hard to see that, especially with the very eye-catching dark green curves).*
- Third, unless $k_{MD} = 0$, the production rate of the authors’ proposed model (terminal differentiation; Fig 1C,F,I,L) is lower than the regular differentiation model (Fig 1B,E,H,K). This seems to be a major caveat of the proposed strategy, as the “defectors” take over more quickly and significantly in the absence of the “cheaters”. Especially since the authors state that “While eliminating this mutation may not be possible,” I believe that Fig 1 needs to be improved to highlight the significance and importance of their proposed system.*
- ▶ In responding, we will use the language non-producers in place of cheaters, and non-differentiators in place of defectors. As you point out in Figure 1B and 1C, there are two cells that are labeled as “cheaters”. We now recognize that this is not accurate and is misleading when we instead use the terminology “non-producers”. The top right cell that was labeled “cheater” is a progenitor cell that has incurred a burden mutation, while the bottom right cell that was labeled “cheater” is a differentiated cell that has incurred a burden mutation. It is therefore accurate to call only the latter a non-producer, and we have changed this in the figures and text. We instead distinguish progenitor cells that have incurred burden mutations as “progenitors*”.
- The strategy of terminal differentiation we demonstrate here prevents burden mutations (which in the context of differentiated cells yield non-producers) from expanding in the population. First of all, having the function activated by differentiation prevents burden mutations from expanding in the progenitor cell population because they do affect the growth rate of the cells while the function is not expressed. Secondly, in terminal differentiation the expansion of non-producer cells is prevented by limiting the growth of **all** differentiated cells (indicated by red X on producers and non-producers). In Figure 1, our primary goals were to demonstrate (1) with non-terminal differentiation, both burden mutations and differentiation mutations both are selected for and expand (Figure 1B, E, H, K); (2) with terminal differentiation, only differentiation mutations, but **not** burden mutations, are selected for and expand in the population (Figure 1C, F, I, L); and (3) reducing the rate of the differentiation mutation uniquely improves the performance of terminal differentiation because it is susceptible **only** to this mutation. This is critical because **differentiation mutations are not specific to the engineered function of interest**, and therefore improving the circuit by reducing the rate/probability of differentiation mutations improves the evolutionary stability of **any arbitrary function**. Terminal differentiation creates a singular weak point that is the category of mutations we describe as differentiation mutations, and addressing this weak point by reducing the rate/probability of these mutations increases the evolutionary stability of an arbitrary function.

As you point out, Figure 1 in the original form is not sufficiently clear, and does not effectively communicate these points. We have modified the color scheme used to make it easier to distinguish what is being plotted in Figure 1D-F, with the colors matching those used in the revised cartoons in Figure 1A-C. As well, Figure 1G-H now clearly show the robustness of terminal differentiation to burden for both total production (G) and duration of function (H). We also show that reducing the differentiation mutation rate uniquely improves total production and duration of function for terminal differentiation. However we have chosen to remove the case of this rate being 0, and instead use a third rate of 10^{-18} (which would be approximately equivalent to three mutations of rate 10^{-6} with deterministic modeling). Showing several rates of the differentiation mutation serves to demonstrate that each reduction of this rate improves evolutionary stability. While it is indeed true that driving this rate to zero will fully prevent loss of function from evolution (though still being susceptible to drift by the linear accumulation of mutations over time that are not being selected), we are no longer emphasizing this point.

- 6) *When first introducing “bacteriophage serine integrases” in the results section, it would be nice to indicate that it’s called “Bxb1” since the figure uses this abbreviation. It may also be helpful to use annotations from the figure in the text to help guide the readers.*
 - ▶ We have clarified this in the manuscript and clearly say that Bxb1 is the bacteriophage serine integrase that is used in this study.
- 7) *The authors should take more time to explain the basic functionality of their platform before jumping into the more complicated, long-term data in the main text.*
 - ▶ In addressing your comment #3, we have added more detail regarding the functionality of the circuits we describe that addresses this comment and adds clarity to the manuscript.
- 8) *In Figure 2, the authors aim to demonstrate the evolutionary stability of their circuits compared to the naïve and non-terminal differentiation counterparts. However, Fig 2 does not convey this message very effectively. For instance, the authors use the cumulative (total) sfGFP as their readout for evolutionary stability, but the naïve version circuit outperforms their proposed circuit at a lower burden based on this metric. Instead, it may be helpful to show the change in sfGFP intensity (or at least highlight the notion of change in production rate), because the authors can reason that the slowdown of sfGFP production may signal when the system has broken to function properly.*
 - ▶ To more clearly show the change in production over the course of the 16 plate generations, we instead plot the endpoint GFP fluorescence in Figure 2A-C. and Figure S11A-C. With this we see clearly that lower differentiation rates allow for longer duration of GFP production though at a lower level, and that terminal differentiation allows GFP production to be maintained at a higher level for longer with the higher differentiation rates (20 and 30 μ M salicylate) where loss of GFP expression is more apparent during this experiment. In both figures we still plot the total GFP production achieved in panel D.
- 9) *The authors argue that the reduced productivity (Fig 2,3) is attributed to broken circuits. The sequencing information on the broken vs. intact circuits at the end of the long term would be compelling to support their argument, especially because the authors computationally model plasmid mutation as a major source of circuit instability in Fig 3. Their*

experimental data (Fig 3F) does not encompass the sequence-level validation of circuit stability. The authors should also examine the fraction of intact vs. cheater populations to validate the evolutionary stability of their engineered population.

Such validation is crucial the authors add chloramphenicol (cm) to select for the R6K plasmid during the entire course of their long-term experiments, whereas the differentiation eliminates the R6K-free population. These opposing two forces can cause severe stress on the cells that they are more prone to break.

Moreover, in Figure 3, the authors specifically investigate on the effect of plasmid mutation using their mathematical model, but in Figure 2, it is ambiguous if they only consider the genome-integrated components when they refer to the evolutionary stability of the “circuit”. Without explicitly examining where in the circuit was compromised, it is inconclusive to claim what were the contributing factors to the observation described in Fig 2 and 3.

- ▶ We appreciate this critique, and examining how each circuit fails is an important validation of this strategy. However, your statement that selecting for the R6K plasmid with chloramphenicol while differentiation results in loss of the R6K plasmid “can cause severe stress on the cells that they are more prone to break” is not entirely accurate, and does not seem to capture the motivation of the terminal differentiation strategy. Without chloramphenicol selection, mutations which disrupt the expression of the engineered function (here T7 RNAP-driven GFP) provide a selective advantage (in the differentiated cell population) and expand in the population after they inevitably occur. The purpose of chloramphenicol selection is to eliminate the opportunity for these burden mutations to expand in the population by limiting the growth of the population of cells in which these mutations provide a selective advantage: Differentiated producers. This chloramphenicol selection does not change the likelihood of mutations occurring, but it does change what types of mutations provide a selective advantage. With terminal differentiation, we expect mutations which disrupt differentiation to be the cause of circuit failure, while for non-terminal differentiation both differentiation mutations and burden mutations can be the cause of circuit failure. From intuition and modeling, we demonstrate what types of mutations and phenomenon disrupt the performance of (1) naïve expression, (2) non-terminal differentiation in which the growth of differentiated cells is not limited, and (3) terminal differentiation in which the growth of differentiated cells is limited. We summarize this below:

- Naïve expression
 - Mutations
 - Genomic mutations which disrupt the expression of T7RNAP
 - Plasmid mutations on the ColE1 plasmid which decrease burden by reducing T7RNAP driven expression.
 - In Figure 1 we do not distinguish between these types of mutations, which are both are burden mutations.
 - In Figure 3C (and Figures S19-S21) in modeling naïve expression, we separately consider genomic mutations that disrupt T7RNAP, and plasmid mutations that disrupt T7RNAP-driven expression. We demonstrate here with stochastic simulations that naïve expression is negatively impacted by increasing plasmid mutation rates, and that redundant naïve expression with

two copies of T7RNAP is more impacted than with a single genomic copy of T7RNAP.

- Non-mutation factors
 - Complete or partial loss of the ColE1 plasmid reduces burden, and in our modeling we demonstrate that communal antibiotic degradation can allow cells that have lost the ColE1 plasmid to grow despite not having the resistant gene.
 - In Figure 3C (and Figures S19-S21) we explore this with modeling, and show that plasmid loss combined with communal antibiotic resistance negatively impacts the production achieved by naïve expression, and that is most pronounced for redundant naïve expression, and for burdens that are high (50%) but not low (10%) or extremely high (10%).
 - In Figure S6 we show that communal antibiotic degradation is readily observed with AmpR as the resistance marker on the ColE1 plasmid, but not with KanR on the ColE1 plasmid. In Figure 3F and Figure S11, we demonstrate that having AmpR on the ColE1 plasmid, allowing for this communal antibiotic degradation, has a striking negative impact on the performance of redundant naïve expression at both burden levels tested, strongly indicating this is at play.
- Differentiation and Terminal differentiation
 - Mutations
 - Both differentiation and terminal differentiation are susceptible to genomic mutations which disrupt differentiation, through (1) disrupting Bxb1 integrase expression, or (2) destroying the integrase attachment sites to prevent recombination.
 - Differentiation but not terminal differentiation is susceptible to genomic mutations which disrupt the expression of T7 RNAP in differentiated cells.
 - Differentiation but not terminal differentiation is susceptible to plasmid mutations on the ColE1 plasmid which decrease burden by reducing T7RNAP driven expression.
 - In Figure 1 and Figure S1 we demonstrate differentiation is susceptible to both burden (not distinguishing between plasmid and genomic mutations) and differentiation (not distinguishing between mutations which disrupt Bxb1 expression or disrupt recombination between integrase attachment sites) mutations, while terminal differentiation is only susceptible to differentiation mutations.
 - In Figure 3D-E we separately consider mutations disrupting integrase expression, mutations disrupting integrase recombination, mutations disrupting T7 RNAP expression, and plasmid mutations. We explicitly show here that the performance of differentiation, but not terminal differentiation, is negatively affected by plasmid mutations. As well, we show that this negative affect is greater for 2x than 1x differentiation.

- Non-mutation factors
 - In Figure 3C (and Figures S22-S27) we demonstrate computationally with stochastic simulations that the performance of terminal differentiation is not impacted by plasmid loss, while it is negatively impacted for differentiation. We specifically show that 2x differentiation is more severely impacted than 1x differentiation, and that this affect is burden and differentiation rate dependent. Specifically, it is more pronounce at higher burden, and is less pronounced at low differentiation rates.
 - Figure 3F (full data in Figure S11) is a convincing experimental demonstration of this. We show that 2x differentiation (without chloramphenicol selection) performs worse when AmpR is used as the selectable marker on the ColE1 plasmid than with KanR, and this is most apparent in the higher burden condition, and increasingly so with higher differentiation rates. In contrast, as we expected from modeling, 2x terminal differentiation was not negatively impacted by the use of AmpR.

To further address your comments and characterize the mechanisms which caused production of GFP to decrease or cease during the experiment, we performed selective plating assays from glycerol stocks saved after the last plate growth, and used Nanopore sequencing to identify causal mutations in a sample of colonies. We chose the higher burden (50 μ M IPTG) and highest differentiation rate (30 μ M salicylate) for this characterization as it was the condition which displayed the largest decrease in GFP production during the experiment. Select dilutions were plated from three of the eight replicates each for 1x and 2x naïve, and 1x and 2x differentiation and terminal differentiation, and 6 colonies total for each strain across replicates were analyzed by sequencing. Sequences for the region of the ColE1 plasmid encoding T7 RNAP-driven GFP and all genomic integrations were obtained, with few exceptions due to failed PCR or insufficient sequencing reads. Naïve 1x and 2x had <1% producers in all replicates as inferred through the fraction of GFP+ colonies. For 1x naïve, sequencing identified causal mutations in the coding sequence of T7 RNAP in all six colonies, with 3 unique frameshift insertion mutations and 3 unique nonsense mutations, and no mutations were observed in the ColE1 plasmid. For 2x naïve, however, no mutations were observed in the T7 RNAP coding sequence, but instead mutations in the T7 promoter on the ColE1 plasmid were present in 4 of the 6 colonies. Mutations in the T7 promoter highlight that contribution of transcriptional burden in this system and the power of random plasmid partitioning in accelerating fixation of mutations. As well, that we see these promoter mutations in 2x naïve but not 1x naïve suggests the aggregate rate of generating and enriching for plasmid mutations through random plasmid partitioning is lower than the rate of genomic mutations which disrupt T7 RNAP expression, but higher than the rate of generating two such mutations. As we discuss the mutations identified in the differentiation and terminal differentiation circuits, we note that no mutations were observed on the ColE1 plasmid apart from those observed with 2x naïve expression.

Figure S29: Characterization of circuit failure mechanism with selective plating and sequencing. **(A)** Serial dilutions of 3 of the 8 replicates from plate generation 16 glycerol stocks from 1x/2x naive high burden (50 μ M IPTG), and 1x/2x differentiation/terminal differentiation high burden/high differentiation rate (50 μ M IPTG/30 μ M salicylate) were plated. Plates were imaged, and GFP+ and GFP- colonies were enumerated for naïve 1x/2x, and RFP(+/-)/GFP(+/-) colonies enumerated for diff./term. diff. 6 colonies were sampled for each strain for colony PCR with barcoded primers, and analysis with Nanopore sequencing **(B)** Fraction of colonies for each category plotted as box plots, with individual replicates included as circles. For differentiation/terminal differentiation, differentiators/producers (blue) are determined

by number of GFP+ colonies on LB + Kan/Las/Sal, non-differentiators (yellow) by number of RFP+/GFP- colonies on LB + Kan/Chlor/Las/Sal, and non-producers/non-differentiators (green) that have lost the R6K plasmid by the number of non-fluorescent colonies on LB + Kan/Las/Sal. For naïve, producer fraction (green) was determined by the number of GFP+ colonies, and non-producer fraction (green) by the number of GFP- colonies. **(C)** Normal differentiation excises the *pir* (or for 2x differentiation an intein-tagged split-*pir* fragment), activating T7 RNAP expression and ablating π -protein expression. Mutations observed in 1x/2x differentiation/terminal differentiation were frequently the result of a recombination error that inverted rather than excised the intervening sequence, resulting in catalytically dead attB/attP sites (intact attB/attP sites are black/white triangles, half black/white shapes reflect inversion). Other mutations observed are described in supplementary file *endpoint_plating_sequencing.xlsx*. **(D)** Non-sense mutations and frameshift insertion mutations were identified for 1x naïve that disrupt the expression of T7 RNAP. **(E)** Mutations in the T7 promoter, but not mutations in the T7 RNAP coding sequence were identified for 2x naïve.

For 1x differentiation, 68-77% of colonies were non-differentiators, 23-32% were non-producers or non-differentiators that had lost the R6K plasmid, and none were producers or functional differentiators (mScarlet+/GFP-, -/-, and GFP+, respectively, when plated on LB + Kan/Las/Sal). For terminal differentiation, two replicates had 100% non-differentiators, and one replicate had ~99.8% non-differentiators and ~0.2% differentiators/producers. From our intuition and modeling, we know that while non-producers and non-differentiators have selective advantage with the differentiation architecture, only non-differentiators have selective advantage with terminal differentiation. The results of the plating assay reflect this, though the high abundance of non-differentiators in non-terminal differentiation suggests that differentiation mutations are more frequent. We sequenced 3 mScarlet+ non-differentiators, and 3 mScarlet-/GFP- colonies. All 3 mScarlet+ non-differentiators had identical inversions, strongly suggesting that Bxb1 had catalyzed an inversion between the attB and attP sites rather than an excision. The sequence resulting from this recombination does not contain functional attB or attP sites, but does retain the capacity for π -protein expression. Of the three mScarlet- colonies, one contained a correctly recombined cassette that had a nonsense mutation in T7 RNAP (W221*), one had an inverted cassette as described with intact *pir*, and one had a large deletion encompassing the attP right half through a portion of NahRAM, also with intact *pir*. These data demonstrate that loss of the R6K plasmid occurs without loss of π -protein expression, and erroneous recombination by Bxb1 which results in a sequence that is inert to recombination is likely more frequent than errors of replication having the same affect or solely disrupting the expression of T7 RNAP.

For 2x differentiation, ~36%, ~17%, and ~1% of colonies were differentiators/producers; ~64%, ~83%, and ~99% were non-producers or non-differentiators that had lost the R6K plasmid; and ~0.1%, <0.1%, and <<0.1% were non-differentiators. For 2x terminal differentiation, ~74%, ~27%, and ~26% were differentiators/producers; ~21%, ~49%, ~71% were non-differentiators (mScarlet+); and ~6%, ~24%, and ~4% were non-producers or non-differentiators that had lost the R6K plasmid. As mScarlet-/GFP- colonies were only observed for 2x terminal differentiation when plating on media lacking chloramphenicol, and no such colonies were observed for 1x terminal differentiation, this suggests that the R6K plasmid is more susceptible to loss with the split- π protein design when there is not selection. Though non-differentiators were rare in the non-terminal differentiation condition, we sequenced 3 non-differentiators (mScarlet+) and 3 non-producers (mScarlet-/GFP-). In the non-producers, one colony had an inversion at the H locus as

described previously with an intact *cfaC-pirR* fusion protein, and the cassette at the T locus had an inversion involving the integrase attachment sites but disrupting the *pirL-cfaN* fusion, ablating functional π -protein expression. The two additional colonies had this same T locus mutation, but high-quality sequences were not obtained for the H locus. For the non-differentiators sequenced, two had inversions maintaining expression of the split π -protein (one with failed sequencing at the H locus), and one replicate had matching 108bp deletions in both copies of the Bxb1 integrase, and both cassettes had not been recombined and did not have any mutations. For 2x terminal differentiation, we sequenced 2 colonies which were GFP+/mScarlet+, both of which had an inversion in the T cassette which maintained the intact *pirL-cfaN* fusion, and a correctly recombined H cassette with no mutations in the T7 RNAP coding sequence. The four non-differentiator colonies (GFP-/mScarlet+) colonies all had mutations which disrupted one or both integrase attachment sites but left π -protein expression intact. Simple inversions resulting from erroneous recombination were the most common, but a large deletion from the attP right half through a portion of T7 RNAP, and inversions involving a partial duplication of *pirL-cfaN* were also observed.

As evidenced through the sequencing of 2x terminal differentiation colonies identifying cells in which 1 cassette had incurred a differentiation mutation preventing its recombination, it is apparent these mutations are selected sequentially. We speculate that one such mutation reduces the rate of differentiation of the cell, and thereby provides a selective advantage. It is also apparent that errors in Bxb1-recombination are frequently the cause of mutations in both differentiation and terminal differentiation, and because these mutations are the most commonly observed they likely occur at a higher rate than mutations due errors in DNA replication. This is also supported by 2x differentiation (-chlor) underperforming 2x naïve in the low burden condition (Figure 2D), as without differentiation mutations, 2x differentiation with high differentiation rates should be comparable to 2x naïve expression.

- 10) *The lower burden naïve population seems to do better or just as good in GFP production compared to their proposed architecture unless the final product is especially detrimental to the host. Perhaps the authors should focus and elaborate more on the content shown in Figure 4 to better demonstrate their system's significance.*
 - ▶ First we would like to clarify that the 2x naïve circuit, but not the 1x naïve circuit, performs better in terms of total production achieved. Second, we fully appreciate that terminal differentiation as a strategy is useful for improving the evolutionary stability of specifically highly burdensome functions. The comparison of lower burden and higher burden conditions both computationally and experimentally serve explicitly to make this point. Our intention was to make this facet of the terminal differentiation strategy clear, and in revising the manuscript we made efforts to emphasize this point further.
- 11) *Figure 4A should be plotted in bar graph (y axis representing CFU). The two axes should be flipped in Figure 4B to match a uniform style throughout the text.*
 - ▶ We have changed this plot to address your comment.

Reviewers' Comments:

Reviewer #1:

Remarks to the Author:

The authors have addressed all of my previous comments. The updated terminology, using non-producers and non-differentiators, makes the states of their system much easier to follow. I appreciate how they also added some more background and introductory text explaining the rationale for their study. The additional experiments further characterizing the growth and differentiation rates of the different circuits answered my questions. The new sequencing data characterizing how mutations "broke" various circuits is also a nice result to include. Few synthetic biology systems can be said to have been modeled and characterized as carefully as these very interesting differentiation circuits.

Reviewer #2:

Remarks to the Author:

The authors have made substantial revisions to the manuscript to address reviewers' comments. From the response letter, the issues I raised are largely addressed and I now have a clearer understanding of the circuit design strategy. As mentioned in my original comments, the core idea to me was very clever and I think I have understood it.

However, I think the manuscript continues to suffer from poor readability. I continue to struggle to go into the depth of the manuscript due to this issue. There are multiple typos in the manuscript, some of which were introduced in the revision. For example, now some of the figure panels (Figure 1I, J) do not match the text description.

Figure 1 is critical to convey the basic design principle but the description is difficult to follow. I think it's probably better to present the circuit operation (and explain it in more detail) in Figures 1I & J first before describing the simulation results. My understanding of the circuit operation was primarily built upon the circuit diagrams, which was further clarified in the summary of different failure modes, described in the response letter.